# Numerical Differentiation and Its Impact on Uncertainty in Learned Dynamical Systems

**Maria Khilchuk,Ilya Markov & Alexander Hvatov**
NSS Lab
ITMO University
Saint-Peresburg, 197101, Russia
{mdkhilchuk,iomarkov,alex_hvatov}@itmo.ru

## Abstract

This work examines how different differentiation techniques impact the discovery of differential equations from data. Since real-world measurements are often noisy, accurately computing derivatives is crucial for reliable algorithm performance. We explore alternatives to finite difference methods, which are prone to instability and amplify data errors. Our study considers four approaches: Savitzky-Golay filtering, spectral differentiation using neural networks, and derivative regularization strategies. By assessing their suitability for realistic scenarios and their influence on equation discovery convergence, we provide insights into enhancing the robustness of data-driven modeling.

## 1 Introduction

Four critical components define the framework of any machine learning model: the architecture, the parameters, the features, and the objective function. Similarly, modern approaches to differential equation discovery treat differential equations (DEs) as machine learning models. This parallel raises key questions about assessing the quality of identified DEs and the associated uncertainties, leveraging established evaluation techniques from machine learning and sensitivity analysis (SA).

The architecture of a machine learning model determines its structure and complexity, encompassing the design choices for machine learning models and the basis functions selected for representation. Parameter uncertainty refers to variability in model parameters, often due to noisy data or suboptimal training, while features represent the input data or derived variables essential to model performance. Finally, the objective function defines the training criterion by minimizing a loss metric like mean squared error, cross-entropy, or physics-informed residuals.

Uncertainty can be assessed for each component:

`Architectural uncertainty` is typically assessed through techniques such as pruning Blalock et al. (2020) or ensemble methods Lakshminarayanan et al. (2017), which quantify the robustness to structural variations.

`Parameter uncertainty` is often analyzed via sensitivity analysis (SA). Local SA methods, such as the one-at-a-time (OAT) approach Hamby (1994), assess the effect of small perturbations in individual inputs while keeping others constant. However, these methods are limited in capturing non-linearities and interactions. In contrast, global SA methods, like Sobol indices Soboĺ (1993), evaluate the impact of all variations across the parameter space and consider input interactions.

`Feature uncertainty` is managed through preprocessing, data augmentation, sampling, feature engineering, and strategies to handle noisy inputs, such as robust normalization techniques Werner de Vargas et al. (2023).

`The objective function`, although often defined by design, can introduce uncertainties when there is misalignment between the target and the model's capacity Gonzalez & Miikkulainen (2020).

Recent differential equation distribution methods allow for threat differential equations as machine learning methods. Therefore, we could also find analogs to machine learning components. Every equation discovery method aims to find the equation structure — terms that are likely to appear in the governing equation for the data; this structure is closely related to a neural network architecture — it describes how features and layers are interconnected.

The second step is to identify the parameters. The parameters are the coefficients within the DE that frequently correspond to physical properties. They could be referred to as neural network weights (and are basically coefficients of a special type of linear regression).

Advancements in DE discovery techniques have refined these components' uncertainty assessment. For example, parameter uncertainty has been addressed using ensemble-based approaches, such as E-SINDy, which employs term library ensembling to handle parameter robustness Fasel et al. (2022). Structural uncertainty has been explored using methods like multi-objective evolutionary optimization combined with Bayesian networks, as demonstrated by Hvatov & Titov (2023). These approaches allow researchers to quantify structural robustness better and align the identified DEs with physical phenomena.

Unlike traditional machine learning, the objective function in DE discovery is more constrained. It is often defined as the discrepancy of the equation, evaluated either in a strong form (e.g., term-by-term residuals) or in a weak form (e.g., weak formulations like wSINDy Messenger & Bortz (2021)). Solver-based methods are also employed to minimize discrepancies between observed data and solutions generated by the identified DE, such as physics-informed criterion (PIC) and others.

The critical distinction between DE discovery and machine learning lies in the treatment of features. The feature is only observational data—it could be a time series or field (a multi-dimensional tensor that could contain time as one axis). However, to build an equation, we require the differentials with respect to every axis up to the given order. Differentials of the input data are not given in most cases and, thus, must be computed numerically. Thus, the machine learning features are engineered within the algorithm.

Noisy measurements exacerbate the challenge of numerical differentiation, leading to errors in derivative estimates. Stable numerical differentiation techniques, such as finite differences, polynomial interpolation, or machine-learning-based methods Chartrand (2011), have been proposed to address these issues. However, the choice of differentiation method may significantly impact the quality of the DE model discovered. Variations in derivative computation propagate uncertainty into both the estimated parameters and the structural accuracy of the DE.

Despite progress in addressing parameter and structural uncertainties in DE discovery, the impact of differentiation methods on feature uncertainty remains underexplored. This paper **aims** to systematically evaluate how differentiation techniques influence the quality of discovered models, with a particular focus on both parameter and structural accuracy under varying levels of data uncertainty.

**Contribution**

- We describe differentiation as an "feature enginnering" uncertainty source in differential equation discovery
- Experimentally prove the obvious fact that better differentiation quality leads to better discovery, but also the non-obvious fact that different methods should be used for noisy and clean data to achieve better performance
- We compare several frameworks (SINDy and EPDE) to make the results more reliable

**Limitations** Experimentally, we show results only for two frameworks. However, there may be different performances in RL-based equation discovery, such as DISCOVER Du et al. (2024).

**Data and code** are available via the anonymized repository `https://anonymous.4open.science/r/UAI_diff-B53D`

## 2 DIFFERENTIAL EQUATION DISCOVERY BACKGROUND

As noted above, in the differential equation as a machine learning model, we can distinguish the components of such a model: structure/architecture, parameters, features, and target function.

For differential equations discovery, as an input, we have the data placed on a discrete grid $X = \left\{ x^{(i)} = \left( x_1^{(i)}, \ldots x_{\dim}^{(i)} \right) \right\}_{i=1}^{i=N}$, where $N$ is the number of observations and dim is the dimensionality of the problem. We mention a particular case of time series, for which $\dim = 1$ and $X = \{t_j\}_{i=1}^{i=N}$.

It is also assumed that for each point on the grid, there is an associated set of observations $U = \left\{ u^{(i)} = \left( u_1^{(i)}, \ldots, u_L^{(i)} \right) \right\}_{i=1}^{N}$ to define a grid map $u : X \subset \mathbb{R}^{\dim} \to U \subset \mathbb{R}^L$. This grid and observations can be used as input data or features in the machine learning model. It is assumed that $u$ is defined explicitly by the model $M$ which has the form:

$$M(S, P, x) \to u(x) : M\left( S, P, x^{(i)} \right) \to u\left( x_i \right) \sim u^{(i)} \tag{1}$$

In Eq. 1, we define two parts of the model in the form of the equation: the structure $S$ and the parameters $P$. We note that we do not expect either interpolation (case $M\left( S, P, x^{(i)} \right) \to u\left( x_i \right) = u^{(i)}$) or approximation (case $M\left( S, P, x^{(i)} \right) \to u\left( x_i \right) \approx u^{(i)}$). It is assumed that the model $M(S, P, x)$ by itself can be interpreted by an expert and used, for example, to predict the behavior of the system in states that have not yet been observed $\tilde{x}^{(j)}$. In an ideal scenario, the discovery of differential equations enables the extraction of the complete set of underlying equations based on observational data. Unfortunately, in practical situations, we can only approximate the system and obtain a rough estimate.

It is convenient to separate numerical characteristics such as the coefficient of the term, the power of the term, and the order of the derivative into a set of parameters $P$, i.e., to make every node or element of the structure parametrized. The optimization process may be separated for structure $S$, and parameter set $P$ [maybe link here]. The target function occurs when a differential equation is represented as a machine learning model. In most cases, the discrepancy between the target and the calculated values or a solver of differential equations is selected as a target function.

For example, hyperbolic heat equation $\tau \frac{\partial^2 u}{\partial t^2} + \frac{\partial u}{\partial t} = \alpha \nabla^2 u$ with parameters $\tau$ (relaxation time) – parameter representing the time lag required for heat flux to respond to a temperature gradient. It ensures a finite propagation speed of thermal signals (unlike the infinite speed implied by the classical heat equation), $\alpha$ (thermal diffusivity) defined as $\alpha = \frac{k}{\rho c}$, where $k$ is thermal conductivity (material's ability to conduct heat), $\rho$ is density, $c$ is specific heat capacity (energy required to raise temperature per unit mass). The structure of the equation includes all the terms $\frac{\partial^2 u}{\partial t^2}$, $\frac{\partial u}{\partial t}$ and all components of $\nabla^2 = \sum_i \frac{\partial^2 u}{\partial x_i^2}$, i.e. $\frac{\partial^2 u}{\partial x^2}, \frac{\partial^2 u}{\partial y^2}, \frac{\partial^2 u}{\partial z^2}$ if we consider standard Cartesian coordinates of the equation and the operations that are used between them in the final form of the equation.

**Equation discovery problem statement** In the case of differential equations discovery and searching for parameters of the equation, symbolic regression is used. Formally, this machine learning symbolic regression problem can be formulated as follows: the set $A_i = (x_i, y_i)$ is given, $i = \overline{1, N}$ - number of observations, $x_i$ - discrete grid, $y_i$ - data measured at grid points and aligned with the grid. It is known that $y_i$ can be determined through a parameterized model as follows $M(S, P, x_i) \sim y_i$, and there is a loss function $L(M(S, P, x_i), y_i)$. Thus, the optimization problem, which we aim to solve, is

$$S^*, P^* = \underset{S, P}{\operatorname{argmin}} L(M(S, P, x_i), y_i) \tag{2}$$

The methods of equation discovery differ in the way of determining $L(\cdot)$, model $M(S, P, x)$, and the optimization way, i.e., how do we achieve argmin. Below, we briefly outline the main groups of methods.

As a classical algorithm in the area, we consider another algorithm, Sparse Identification of Nonlinear Dynamics (SINDy) Brunton et al. (2016) presented in the form of the PySINDy library, which provides opportunities for finding control equations based on data even in cases of chaotic dynamical systems. This algorithm is based on the idea that we need to find coefficients for known observable

terms, which are often represented as derivatives and have non-linearity in the form of given functions. At the same time, the solution should include as few functions as possible because otherwise, the solution converges worse.

$$\mathbf{\Xi} = \underset{\mathbf{\Xi}'}{\arg\min} \left[ \left\| \mathbf{\Theta}\mathbf{\Xi}' - \dot{\mathbf{X}} \right\|_2 + \alpha \left\| \mathbf{\Xi}' \right\|_1 \right] \tag{3}$$

where $\mathbf{X}$-future measurement data, $\dot{\mathbf{X}}$ is derivative of $\mathbf{X}$, $\mathbf{\Theta}(\mathbf{X})$- library of nonlinear functions generated by $\mathbf{X}$, $\dot{\mathbf{X}} = \mathbf{\Theta}(\mathbf{X}) \cdot \mathbf{\Xi}$- the way to introduce sparse coefficient matrix $\mathbf{\Xi}$ and a regularization parameter $\alpha$ that determines the desired sparsity of the solution Brunton et al. (2016). In our terms, SINDy is the algorithm where structure $S$ is fixed, and we only optimize parameters $S$.

In this regard, the field of methods for searching for differential equations can be much improved; in particular, in addition to regression, evolutionary optimization is actively used, which refers to genetic algorithms in combination with sparse regression, that is realized in EPDE (Evolutionary Partial Differential Equations) framework Maslyaev et al. (2021). In this algorithm, instead of the usual terms," building blocks" are used - these are tokens, a combination of elementary functions from data and from data derivatives. These tokens can include higher-order derivatives, grid functions, and their combination with other elementary functions. All tokens together are formed into a set of tokens $F = \bigcup_i F_i$, where $j$ defines the name of the token family. Next, evolutionary optimization in model training is used for the selected set of tokens.

$$M(C, \mathbf{P}, \bar{x}) = \sum_{i=1}^{i \leq L} c_i * a_i (P_i, \bar{x}) \tag{4}$$

Here, $C = c_i$, $i = \overline{0, L}$ represents the constants before the terms of the equation, $a_i(P_i, \bar{x}) = \prod_{j=1}^{j=N_{\text{tokens}}} f_j \left( p_1^{(i)}, p_2^{(i)}, \ldots, \bar{x} \right)$ denotes the products of tokens $f_j$ from the token families $F$, $P_i = \left\{ p_1^{(i)}, p_2^{(i)}, \ldots \right\}$ represents the parameter set for term $a_i$, and $\mathbf{P}$ represents the parameter multi-index.

As mentioned before, the model $M(S, P, x)$ is assessed by the loss function $L(\cdot)$. In the SINDy case, it is a model discrepancy with respect to the selected term (in SINDy, it is usually $\frac{\partial u}{\partial t}$). In EPDE, one may choose between discrepancy and multi-objective optimization with additional objectives such as complexity. We also may add physics-informed loss, as done in PIC (with SINDy), or replace discrepancy with the difference between the equation solution and data obtained with PINN architecture.

Except for the last case, we require separating data to compute loss, and this is one of the main problems—the solution of the equation is computationally expensive, and if we replace it with a less costly method, we should be able to handle differentiation operations properly.

## 3 DATA DIFFERENTIATION PROBLEM STATEMENT AND PROPOSED METHODS

In what follows, we will primarily discuss reducing random error in the measurements. While other sources of inaccuracies in the data, such as systematic errors, can be significant, they tend to be elusive even though they significantly affect the resulting data-driven equation.

Let us denote the input data for the differential equation discovery algorithm as $u(t, \mathbf{x})$, which is collected as measurements and, in addition to the correct state of the system $\bar{u}(t, \mathbf{x})$ contains noise $n(t, \mathbf{x})$. While we can be sure in the presence of noise in the data, several assumptions can be made about the distribution $F$, to which it belongs. The measurement at the point $(t, \mathbf{x})$ is assumed to be drawn from the Gaussian distribution (Additive White Gaussian Noise, AWGN) with its mean $\bar{u}(t, \mathbf{x})$ - the correct value of the underlying process. In our experiments, we introduce the noise standard deviation $\sigma$ dependent on the variable state $\sigma = \kappa \bar{u}(t, \mathbf{x})$.

$$u(t, \mathbf{x}) = \bar{u}(t, \mathbf{x}) + n(t, \mathbf{x}), \ n(t, \mathbf{x}) \sim F(t, \mathbf{x}) \tag{5}$$

This section is devoted to presenting alternative tools for calculating the derivatives of a modeled function. The baseline approach to numerical differentiation involves finite-difference schemas that employ values of the dependent variable in grid nodes to calculate its derivatives.

$$\frac{\partial u(t, \mathbf{x})}{\partial x_i} \approx \frac{\Delta_{\delta,i} u}{\delta_i} = \frac{u(t, \mathbf{x} + \delta_i) - u(t, \mathbf{x})}{\delta_i}, \tag{6}$$

where the partial derivative of the data-representing function $u(t, \mathbf{x})$ over the $i$-th spatial axis is reconstructed with the values in nodes $(t, \mathbf{x} + \delta_{\mathbf{i}})$ and $(t, \mathbf{x})$ with the finite-difference operator "forward" $\Delta_{\delta,i}$. By $\delta_{\mathbf{i}}$ we denote the vector of increment over the $i$-th axis, $\delta_i^j = 0, i \neq j$, and $\delta_i^i$ - non-zero step of the grid.

With the data contaminated in the manner presented in Eq. 5, it is possible to estimate the quality of derivatives based on the finite differences. Let us assume that the input data on the compact $\Omega$ belong to the Sobolev space $W^{k,p}(\Omega)$ of functions that have their derivatives up to $k$-th order belong to the $L^p(\Omega)$ space (have finite Lebesgue integral): $\overline{u} \in W^{k,p}(\Omega)$. Although we cannot be sure of the same properties of the observation $u$, it can still be attributed to the Lebesgue space with $\infty$-norm: $u \in L^\infty(\Omega)$. In this case, the finite-difference discrepancy can be estimated from the norms in the corresponding spaces:

$$\|\overline{u}'_{x_i} - \frac{\Delta_{\delta,i} u}{2\delta_i}\|_p \leq \|\frac{\Delta_{\delta,i}(u - \overline{u})}{2\delta_i}\|_p$$

$$+ \|\overline{u}'_{x_i} - \frac{\Delta_{\delta,i}\overline{u}}{2\delta_i}\|_p \leq \frac{2\delta_i}{h} + \frac{hC}{2}, \quad (7)$$

whereby $\|\cdot\|_p$ we denote the norm in space $L^p(\Omega)$, and $C$ is the constant obtained from the Taylor series derivation of finite differences: $C \geq \|f"_{x_i x_i}\|_p$. This estimation indicates that the derivatives are sensitive to errors in the measurement. Furthermore, reducing the grid step, usually preferable due to the lower pure numerical error in the finite difference, leads to the magnification of random errors.

The noise influence on the data can be viewed from the point of view of Fourier analysis. The studied process shall not produce high-frequency oscillations or have amplitudes significantly lower than the low-frequency counterparts. If the opposite is true, the data may have aliasing problems, thus limiting the applicability of the frequency-based analysis. These high-frequency components in the DFT (discrete Fourier transform) are linked to the measurement noise or small-scale processes that shall be omitted during the equation construction and filtered out. In what follows, brief notes of applied differentiation methods are presented, with a more detailed and expanded formulation in Appendix A.

- **Filtering-based approaches:** One of the approaches considered in this work involves approximating the input data with the fully connected artificial neural network (ANN). One of the valuable properties of the artificial neural network is that the low-frequency signal in the data is learned first, while further training approximates the high-frequency components Rahaman et al. (2019). Thus, by training an ANN representation of the process, we can obtain its low-frequency approximation, which can be further differentiated with a decreased noise component.

  Savitzky-Golay (SG) filtering, developed in Savitzky & Golay (1964), is a commonly used approach to signal or data filtering, coupled with an opportunity to compute derivatives, involves a least squares-based local fitting of the polynomials to represent the data. For each grid node, the data in its proximity is used to construct a polynomial that can be analytically differentiated.

- **Spectral domain differentiation:** Although the process of differentiation in the spatial domain can be complicated for the data described with an arbitrary function, in the Fourier domain, the derivatives can be estimated on a term-to-term basis Johnson (2011). The discrete Fourier transform (DFT) is the basis of our implementation of spectral domain differentiation. In the spectral domain, integration and differentiation can be maintained

by multiplication of series terms with an appropriate exponential. This leads to low computational costs, especially if the data are located on a uniform grid, thus allowing the use of the Fast Fourier Transform instead of DFT. The Butterworth filter does the signal filtering, which can preserve the signals with frequencies lower than the cutoff frequency while dampening the high-frequency ones.

- **Total variation regularization:** Variational principles provide an alternative method that incorporates inverse problem solution with the regularization of the gradient variation or its higher-order analogs (e.g., Hessian). One of the main advances in this field was made in Chartrand (2011; 2017).

## 4 EXPERIMENTS

To investigate the influence of the above differentiation methods on the discovery of differential equations, we will conduct a series of complex numerical experiments. The search for equations will be carried out using discharged regression and an evolutionary approach.

### 4.1 EXPERIMENTAL SETUP

Several types of partial differential equations were chosen, each with different solutions: analytical (KdV), numerical (Burgers, wave, Laplace), and data modeling behavior in the atmosphere. Also we take benchmark Ross et al. (2023) where the exact equation is not known a priori (it is refered below as pyqg).

The workflow includes selecting and generating data; as noted earlier, it is either obtaining an analytical solution in the form of a matrix of values or finding a solution matrix using numerical methods, setting boundary and initial conditions, where necessary, and choosing constants. After that, all the data obtained are differentiated by the described methods, while the derivatives sought are those that, as is known in advance, occur in the equation. These may be derivatives of the form $\frac{\partial u(t,\mathbf{x})}{\partial x}$, $\frac{\partial u(t,\mathbf{x})}{\partial t}$, $\frac{\partial^2 u(t,\mathbf{x})}{\partial x^2}$, etc. Then, an evolutionary algorithm is applied using the EPDE framework. Data is loaded with the grid and all derivatives, and then we choose a multi-objective mode. The population size is 7 for all equations, and the number of training epochs ranges from 30-80 depending on the complexity of the equation; the maximum number of terms in the equation is 8 for all equations; this is done to obtain greater variability in the equations. Then the algorithm is run; one run gives about 5-7 equations per the Pareto frontier; we make only 50 runs for each equation to avoid too much variation in the data and more accurately estimate the average approximation for all coefficients.

For each data set, a series of experiments was carried out, resulting in boxplots showing the distribution of coefficients in front of the correct terms. The difference between the obtained equations and the true ones was also analyzed using the difference metric—structural hamming distance (SHD).

### 4.2 ORDINARY DIFFERENTIAL EQUATION

As a simple example, we consider second-order ODE in the form

$$mu'' + qu' + ku = 0 \tag{8}$$

with parameters $m = 1$, $q = 0.25$, $k = 3$ and initial conditions $u(0) = 1, u'(0) = 0$. The detailed experimental results are placed in Appendix B with a further discussion in Section 5.

### 4.3 KORTEWEG – DE VRIES EQUATION

The Korteweg-de Vries equation is a partial differential equation $u'_t + u'''_{xxx} + 6uu'_x = 0$, which is one of the few that has analytical one-soliton and two-soliton solutions.

We will study its one-soliton solution, presented in the following form

$$u(x,t) = \frac{2(k^2)}{ch^2(-k(\mathbf{x} - 4(k^2)t))} \tag{9}$$

,where $k = 0.7$ is the constant that determines the velocity of the soliton $4k^2$ and the amplitude $2k^2$. The detailed experimental results are placed in Appendix C with a further discussion in Section 5.

## 4.4 BURGER'S EQUATION

$$u'_t + uu'_x = vu''_{xx} \tag{10}$$

,where $v = 0.05$ is the diffusion coefficient.

The solution was obtained using an implicit numerical scheme for the diffusion term and an explicit numerical scheme for the convective term. An initial condition was set, and the right and left boundaries were fixed at zero. The detailed experimental results are placed in Appendix D with a further discussion in Section 5.

## 4.5 WAVE EQUATION

$$u''_{xx} = c^2 u''_{tt} \tag{11}$$

, where $c = 0.25$ is the propagation speed of the wave. The initial conditions were set as a sinusoidal function; the boundary conditions were fixed at zero, and the finite difference method solved the problem. The detailed experimental results are placed in Appendix E with a further discussion in Section 5.

## 4.6 LAPLACE EQUATION

$$u''_{xx} + u''_{yy} = 0 \tag{12}$$

The Dirichlet boundary conditions were set, and the problem was solved using the finite difference method. The detailed experimental results are placed in Appendix F with a further discussion in Section 5.

## 4.7 QUASIGEOSTROPHIC POTENTIAL VORTICITY EQUATION

Original data was obtained via the pyqg framework for quasigeostrophic modeling. The maximum number of terms for the pyqg case was extended to 15 to capture the complex dynamics of the data. The equations discovered from quasigeostrophic data cannot be effectively characterized using Hamming distance or coefficient distribution, as the governing equations' exact form is unknown a priori. Instead, the most appropriate metric for evaluation is the discrepancy between the original data and the numerical solutions obtained via a differential equation solver based on Physics-Informed Neural Networks (PINNs).

The use of PINNs is necessitated by the high degree of non-linearity in the inferred equations, which renders conventional Finite Element Method (FEM) implementations inadequate for capturing the system's dynamics. Dirichlet boundary conditions were set from data to solve boundary value problems with generated equations. As original data contains stationary potential vorticity field the general form of the governing equation is given by:

$$\mathbf{V_g} \cdot \nabla q = 0 \tag{13}$$

, where $V_g$ represents the geostrophic velocity and q denotes the potential vorticity. Equations discovered:

Via spectral domain differentiation

$$\begin{aligned}
&3.2602 \times 10^{-6} u_{xx} + 0.0067028u + 0.7095u_y \\
&- 0.6485u_y \cos(1.7965y) + 2.5201 \times 10^{-5} uu_{yy} \\
&- 0.01010u_x u - 1.2018 \times 10^{-6} u_{xx}u_{yy} \\
&+ 3.2084 \times 10^{-5} yu_{yy} + 2.4292 \times 10^{-5} u_{xx}u_y \\
&+ 0.1998u_y \sin(2.7363y) - 0.000223 - yu_y = 0
\end{aligned} \tag{14}$$

Via SG filtering

$$0.044994162\,u_x - 5.34527 \times 10^{-5}\,u_{xx}$$
$$- 0.000760196\,u_x u_{yy} + 0.001192827 - u_x u = 0 \tag{15}$$

Presented equations were discovered using Savitzky-Golay (SG) filtering and spectral-domain differentiation methods, respectively, as alternative approaches failed to capture the eddy-driven structure of the derivatives, leading to suboptimal preprocessing. The solutions to these equations similarly exhibit a lack of regions with pronounced eddy behavior, which may indicate a tendency toward identifying broader-scale features in the data. Visual representations of the original data, numerical solutions, and error maps are provided in Appendix G.

These results show that, in real cases, we cannot achieve consistent results for unknown equations and that we require optimizing errors using differentiation methods as a hyperparameter.

## 5 DISCUSSION

During the experiments, we also gather SHD (structural Hamming distance, see Appendix H for detailed results) and differentiation errors (see detailed results in Appendix I) for every field and every equation. The integral characteristics are shown in Tab. 1, Tab. 2, and Tab. 3 for every noise level considered and for both methods (SINDy and EPDE) simultaneously (detailed results are available in appendices for every equation considered).

Table 1: Means of scores, noise=0

| Methods | D. error | Coeff. error | SHD |
|---|---|---|---|
| Gradient | 0.003248 | 0.7309±0.0515 | 2±0.0782 |
| Adaptive | 8035.51 | 0.7354±0.0522 | 2±0.091 |
| Polynomial | 0.9285 | 0.8971±0.0517 | 2±0.13 |
| Spectral | 9988.1043 | 1.1074±0.0461 | 4±0.1309 |
| Inverse | 1041.2036 | 1.0482±0.0518 | 4±0.1929 |
| Total | 1056.3452 | 1.2669±0.0878 | 3±0.1055 |

Table 2: Means of scores, noise=0.5

| Methods | D. error | Coeff. error | SHD |
|---|---|---|---|
| Gradient | 0.00602 | 0.9179±0.0683 | 3±0.1153 |
| Adaptive | 8279.4516 | 0.9989±0.0497 | 4±0.1349 |
| Polynomial | 1358.1108 | 0.9611±0.039 | 3±0.126 |
| Spectral | 10434.3221 | 1.1737±0.0462 | 5±0.1558 |
| Inverse | 1698.2769 | 1.2806±0.0656 | 4±0.1776 |
| Total | 1419.437 | 1.4814±0.1017 | 3±0.0997 |

Table 3: Means of scores, noise=1

| Methods | D. error | Coeff. errors | SHD |
|---|---|---|---|
| Gradient | 0.0168 | 1.0068±0.2449 | 4±0.1482 |
| Adaptive | 9119.1626 | 1.0602±0.0462 | 3±0.131 |
| Polynomial | 5717.7849 | 0.9148±0.1551 | 3±0.1323 |
| Spectral | 11392.3523 | 1.1924±0.0466 | 4±0.1498 |
| Inverse | 10990.5993 | 1.3120±0.0816 | 5±0.1537 |
| Total | 2797.4304 | 1.5737±0.0878 | 3±0.1063 |

As a rule of thumb, a lesser differentiation method leads to lower SHD since, in differential equation discovery, we are mostly interested in the structure, not the proper coefficients. Coefficients could be obtained after discovery using other means, such as different types of regression.

Remarkably, for high noise in Tab. 3, the second-best method provides the best result despite the low gradient method error.

## 6 CONCLUSION

The paper considers another aspect of differential equation discovery as a machine learning method.

The error of the differentiation algorithm as the "feature engineering" method plays a role in the general uncertainty and is often left out of the scope.

The main results are as follows.

- The differentiation is an important part of every equation discovery method
- Best differentiation methods for noise data and clean data are different
- Absolute value of differentiation error is less important – very precise methods give poor discovery results in some cases

We also mention that the conclusion remains the same regardless of the method used, LASSO regression-based SINDy or evolutionary EPDE.

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

# A  DIFFERENTIATION APPROACH FORMULATION

## A.1  SAVITZKY-GOLAY FILTERING

Savitzky-Golay (SG) filtering, developed in Savitzky & Golay (1964), is a commonly used approach to signal or data filtering, coupled with an opportunity to compute derivatives, involves a least squares-based local fitting of the polynomials to represent the data. To the set of data samples along an axis, we introduce the window of (commonly, odd) length $N = 2M + 1$, allowing the construction of series of polynomials $P_0(x), P_1(x), ...$ up to (even) order $n$, $n < N$ to approximate the data in the interior of our domain. With the selection of appropriate window size, from which the function values are used for the approximation, and polynomial order, the overdetermined system is constructed. Its solution provides the polynomial coefficients that represent the smoothed signal, without oscillations, caused by the random error. Even though the boundaries of length $M$ can be processed in a separate way, with the finite-difference schema or by a shifted approximation, the quality of results tend to decrease, thus for the equation discovery only the domain interior shall be used.

During the calculation of the partial derivative $u'_j$ for the sample $u(x_i)$, matching the $x_i$ grid node along the $j$-th axis, we select samples $\mathbf{u}_i = (u_{i-M}, u_{i-M+1}, ... , u_i, ... , u_{i+M})$ in the aforementioned window. Using the corresponding coordinates $\mathbf{y}_i = (x_{i-M}, ... , x_i, ... , x_{i+M})$, we introduce the least-square problem of detecting coefficient vector $\alpha = (\alpha_0, ... , \alpha_{n-1})$ for the series $P_0, ... , P_{n-1}$. The representation of data samples is as follows:

$$u_i = \sum_{k=0}^{n-1} \alpha_k P_k(x_i). \tag{16}$$

$$\alpha = \arg\min_{\alpha'} |\mathbf{u}_i - P\mathbf{y}_i|, \tag{17}$$

where matrix $P$ contains values of the polynomials in the grid nodes.

In our case, we utilize orthogonal Chebyshev polynomials of the first kind, where by $C_m^{2k}$ we denote the number of combination of $2k$ elements from the set of cardinality $m$:

$$T_m(x) = \sum_{k=0}^{\lfloor m/2 \rfloor} C_m^{2k}(x^2 - 1)^k x^{m-2k} \tag{18}$$

Having a series of Chebyshev polynomials with calculated coefficients, differentiation can be held analytically. Using the representation of data as series in 16, we get the derivative as $u'_i = \sum_{k=0}^{n-1} \alpha_k U_k(x_i)$, where $U_k$ is a Chebyshev polynomial of the second kind.

$$U_m(x) = \sum_{k=0}^{\lfloor m/2 \rfloor} C_{m+1}^{2k+1}(x^2 - 1)^k x^{m-2k} \tag{19}$$

Although the provided approach is capable of filtering the data and stably calculating the derivatives, work Schmid et al. (2022) suggests that modification of Savitzky-Golay filtering by adding fitting weights or by implementing other filters, such as Whittaker-Henderson filter, can lead to better results in noise suppression.

## A.2  SPECTRAL DOMAIN DIFFERENTIATION

Although the process of differentiation in the spatial domain can be complicated for the data, described with an arbitrary function, in the Fourier domain the derivatives can be estimated in term-to-term basis Johnson (2011). In general, the series of the derivatives, taken on a term-to-term basis may not converge. However, if we assume that the data represents continuous piecewise smooth function that has piecewise differentiable derivatives, the data can be differentiated term-to-term.

A discrete Fourier transform (DFT) is the basis for our implementation of spectral domain differentiation. Let us examine a case of one-dimensional data, even though the algorithm can operate on multi-dimensional data, with the canonical discrete Fourier transform algorithm replaced by n-dimensional DFT. In data-driven equation discovery problems, one-dimensional data $u(t)$ is viewed from the point of view of samples $u_n = u(nT/N), n = 0, 1, \ldots, N - 1$, where $T$ is the length of time interval and $N$ - the number of samples, and the corresponding coordinates will be $t_n = nT/N, n = 0, 1, \ldots, N - 1$. The Fourier coefficients are denoted as $\hat{u}_k$, and they are calculated as:

$$\hat{u}_k = \frac{1}{N} \sum_{n=0}^{N-1} u_n exp(-2\pi i \frac{nk}{N}). \tag{20}$$

In many cases, the data are provided on the regular (even multi-dimensional) grid, thus to improve the algorithm performance a fast Fourier transform can be used. Due to the lower computational complexity, the increase in performance is substantial. The process of data reconstruction, using the obtained Fourier coefficients, is held with an inverse discrete Fourier transform:

$$u_n = \sum_{k=0}^{N-1} \hat{u}_k exp(2\pi i \frac{nk}{N}). \tag{21}$$

Full term-by-term differentiation is performed in the Fourier domain, and the derivatives values are computed by the inverse DFT. For example, an expression for the first-order derivative has form, as in Eq. 22.

$$u'(t_k) = \sum_{0 < k < \frac{N-1}{2}} \frac{2\pi i}{T} k \left( \hat{u}_n exp(2\pi i \frac{nk}{N}) - \hat{u}_{N-k} exp(-2\pi i \frac{nk}{N}) \right). \tag{22}$$

Filtering with the desired properties can be done with low-pass filters that pass signals with lower frequencies, while dampen the high-frequency ones. Butterworth filter is a representative of such tools, and is flat for the passband (the frequencies that we do not want to penalize). The latter property prevents distortion of the modeled process by introducing factors, close to 1, to the low-frequency Fourier components. The penalizing factor is introduced with the expression eq. 23:

$$G(\omega) = \frac{1}{1 + (\omega/\omega_{cutoff})^{2s}}, \tag{23}$$

where $\omega$ is the frequency, $\omega_{cutoff}$ is the cutoff frequency, indicating the boundary frequency, from which the damping begins, and $s$ is the filter steepness parameter. The resulting expression is obtained with the introduction of penalizing factors $G(\omega) = G(k/N)$ into the series, representing derivatives:

$$u'(t_k) = \sum_{0 < k < \frac{N-1}{2}} G(k/N) \frac{2\pi i}{T} k \left( \hat{u}_n exp(2\pi i \frac{nk}{N}) - \hat{u}_{N-k} exp(-2\pi i \frac{nk}{N}) \right) \tag{24}$$

The derivative of the higher orders can be calculated recursively from the lower order ones with the same filtering-based differentiation procedures, or, preferably, by the further multiplication with the integrating coefficient and IDFT.

## A.3 TOTAL VARIATION REGULARIZATION

Variational principles provide an alternative method that incorporates inverse problem solution with the regularization of the variation of the gradient or its higher order analogues (e.g. Hessian). Rudin-Osher-Fatemi model Rudin et al. (1992) in its discrete formulation can be represented by the optimization problem of minimizing functional 25.

$$|D(\nabla \cdot u)|_1 + \frac{\mu}{2}|K(\nabla \cdot u) - u|_2^2 \longrightarrow \min_{u}, \tag{25}$$

where $\nabla \cdot u = (\frac{\partial u}{\partial t}, \frac{\partial u}{\partial x_1}, \ ...)$ is the gradient of the data field and $K$ and $D = (D_t, D_{x_1}, D_{x_2}, \ ...)$ represent discrete integration operators onf differentiation. Regularization of gradient variation is maintained with term $|D(\nabla \cdot u)|_1 = \sum_\Omega \sqrt{\sum_{i,\,j} \frac{\partial^2 u)}{\partial x_i \partial x_j}}.$

Chartrand (2011; 2017)

Although there are multiple approaches to the solution of the problem, we employ an approach, proposed in articles Chartrand (2011; 2017), that is designed for a function of one variable. While this approach can be generalized to the problems of higher dimensionality, the computational costs associated with the optimization limit the method's applicability to large datasets. To perform the functional optimization required in Eq. 25, the corresponding Euler-Lagrange equation has to be formed and solved.

## B  ODE EQUATION COEFFICIENTS AND DIFFERENTIATION ERRORS

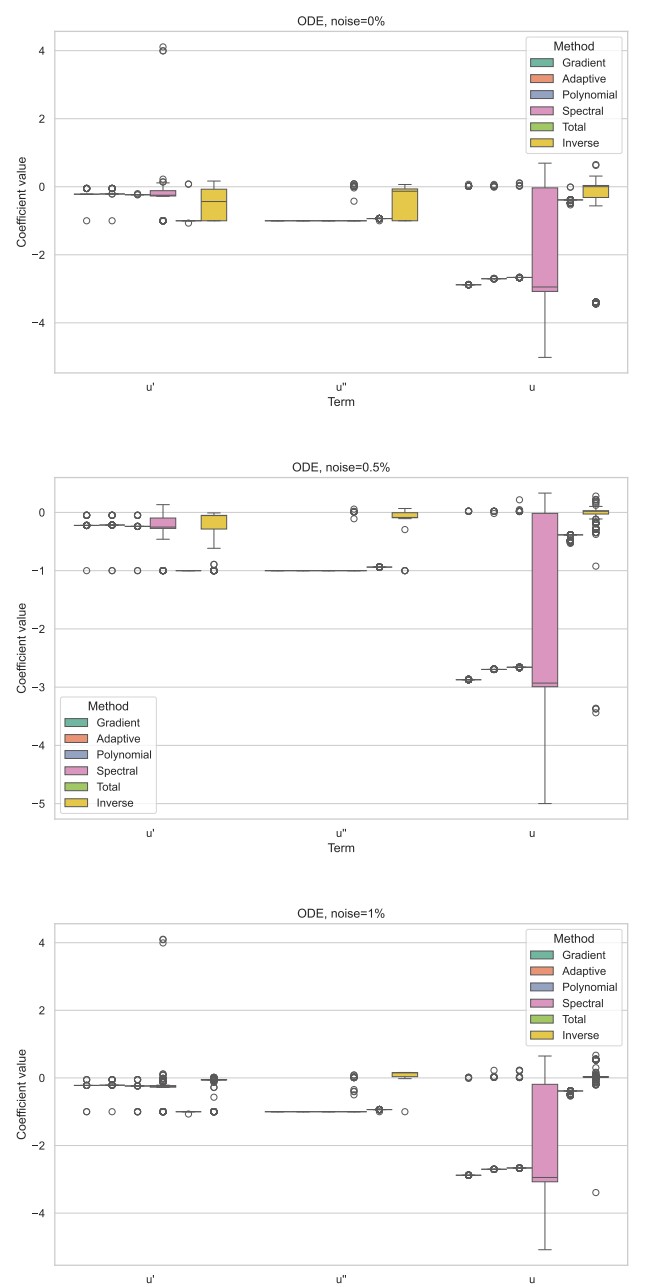

Figure 1: Distribution of coefficients values for different noise level

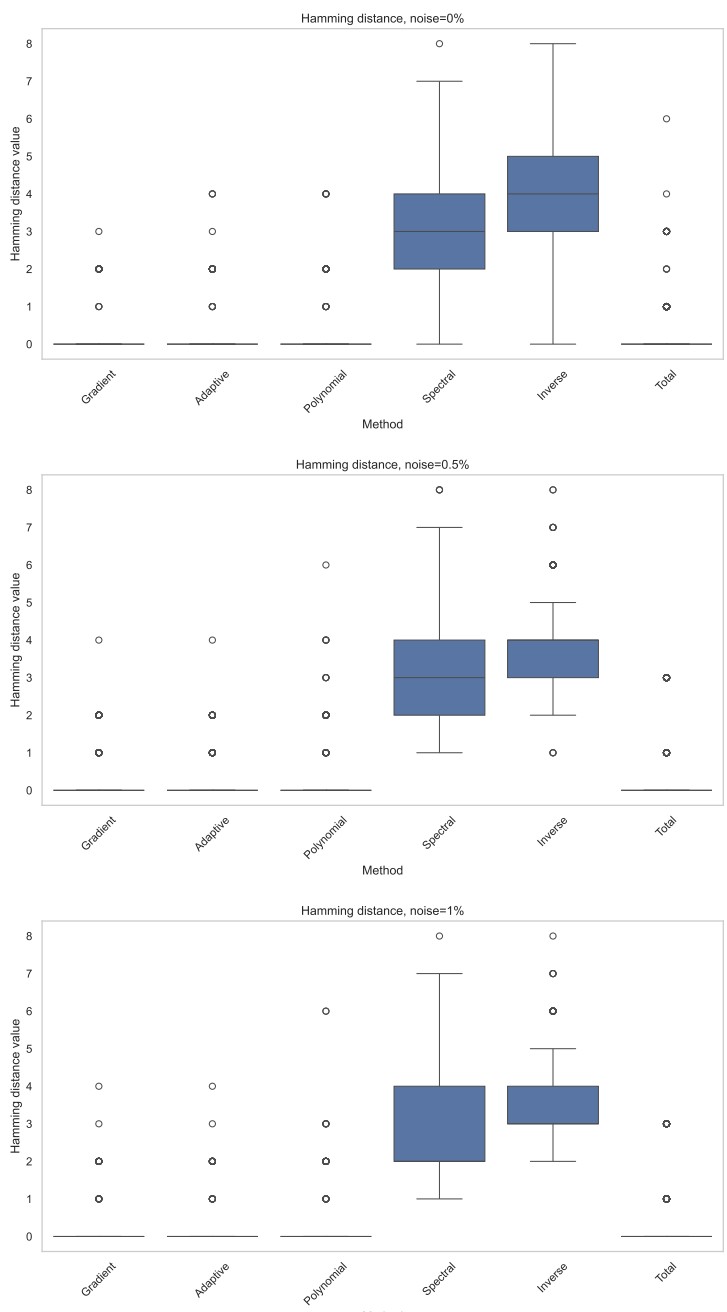

Figure 2: Distribution of coefficients values for different noise level

Table 4: Coefficients values calculated with EPDE, noise = 0

| Methods/Terms | u | u' | u" |
|---|---|---|---|
| Gradient | $0.0219 \pm 0.0082$ | $-0.2156 \pm 0.0048$ | -1 |
| Adaptive | $0.0197 \pm 0.0077$ | $-0.2086 \pm 0.0053$ | -1 |
| Polynomial | $0.0541 \pm 0.0452$ | $-0.2348 \pm 0.0002$ | -1 |
| Spectral | $-0.0312 \pm 0.0412$ | $-0.3008 \pm 0.0392$ | $-0.8997 \pm 0.0460$ |
| Inverse | $-0.0142 \pm 0.0282$ | $-0.5213 \pm 0.0632$ | $-0.5069 \pm 0.1026$ |
| Total | $-0.3901 \pm 0.0032$ | $-0.9952 \pm 0.0068$ | $-0.9353 \pm 0.0003$ |
| Ground truth | 3 | 0.25 | 1 |

Table 5: Coefficients values calculated with EPDE, noise = 0.5

| Methods/Terms | u | u' | u" |
|---|---|---|---|
| Gradient | 0.0205 ± 0.0000 | -0.2222 ± 0.0043 | -1 |
| Adaptive | 0.0152 ± 0.0112 | -0.2173 ± 0.0055 | -1 |
| Polynomial | 0.0363 ± 0.0331 | -0.2412 ± 0.0058 | -1 |
| Spectral | -0.0288 ± 0.0241 | -0.2694 ± 0.0375 | -0.9048 ± 0.0480 |
| Inverse | -0.0255 ± 0.0177 | -0.2840 ± 0.0533 | -0.1348 ± 0.0804 |
| Total | -0.3937 ± 0.0027 | -1 | -0.9387 |
| Ground truth | 3 | 0.25 | 1 |

Table 6: Coefficients values calculated with EPDE, noise = 1

| Methods/Terms | u | u' | u" |
|---|---|---|---|
| Gradient | 0.0098 ± 0.0251 | -0.2248 ± 0.0053 | -1 |
| Adaptive | 0.0444 ± 0.0523 | -0.2149 ± 0.0040 | -1 |
| Polynomial | 0.0983 ± 0.0748 | -0.2455 ± 0.0082 | -1 |
| Spectral | -0.0651 ± 0.0643 | -0.3254 ± 0.0391 | -0.9172 ± 0.0400 |
| Inverse | 0.0639 ± 0.0191 | -0.1889 ± 0.0410 | 0.0789 ± 0.0746 |
| Total | -0.3949 ± 0.0027 | -1.0002 ± 0.0003 | -0.9371 ± 0.0003 |
| Ground truth | 3 | 0.25 | 1 |

Table 7: Coefficients values calculated with SINDy, noise =0

| Methods/Terms | u | u' | u" |
|---|---|---|---|
| Gradient | 2.845 | 0.208 | 1 |
| Adaptive | 2.385 | 0.249 | 1 |
| Polynomial | 2.874 | 0.193 | 1 |
| Spectral | 3.199 | - | 1 |
| Inverse | 2.732 | 0.264 | 1 |
| Total | 0.413 | 1.070 | 1 |
| Ground truth | 3 | 0.25 | 1 |

Table 8: Coefficients values calculated with SINDy, noise =0.5

| Methods/Terms | u | u' | u" |
|---|---|---|---|
| Gradient | 2.824 | 0.212 | 1 |
| Adaptive | 2.368 | 0.253 | 1 |
| Polynomial | 2.854 | 0.206 | 1 |
| Spectral | 3.180 | - | 1 |
| Inverse | 3.697 | 0.376 | 1 |
| Total | 0.409 | 1.066 | 1 |
| Ground truth | 3 | 0.25 | 1 |

Table 9: Coefficients values calculated with SINDy, noise =1

| Methods/Terms | u | u' | u" |
|---|---|---|---|
| Gradient | 2.754 | 0.212 | 1 |
| Adaptive | 2.353 | 0.256 | 1 |
| Polynomial | 2.785 | 0.176 | 1 |
| Spectral | 3.197 | - | 1 |
| Inverse | 3.240 | 0.268 | 1 |
| Total | 0.414 | 1.072 | 1 |
| Ground truth | 3 | 0.25 | 1 |

# C  KDV EQUATION COEFFICIENTS AND DIFFERENTIATION ERRORS

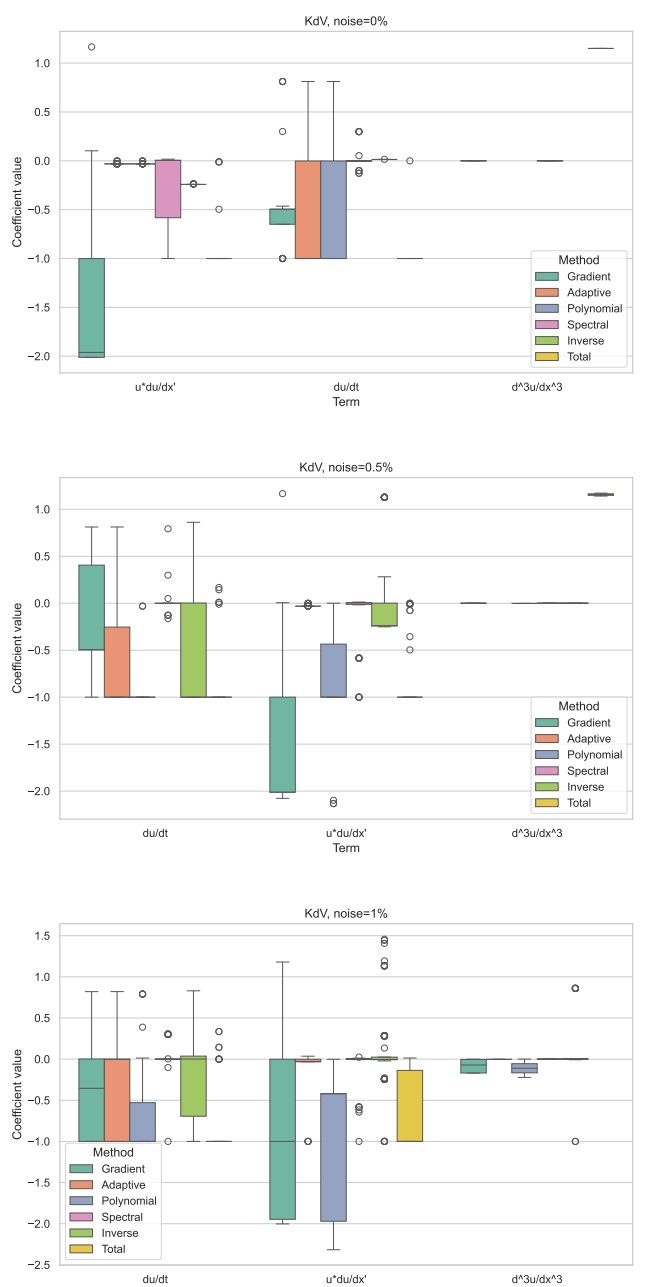

Figure 3: Distribution of coefficients values for different noise level

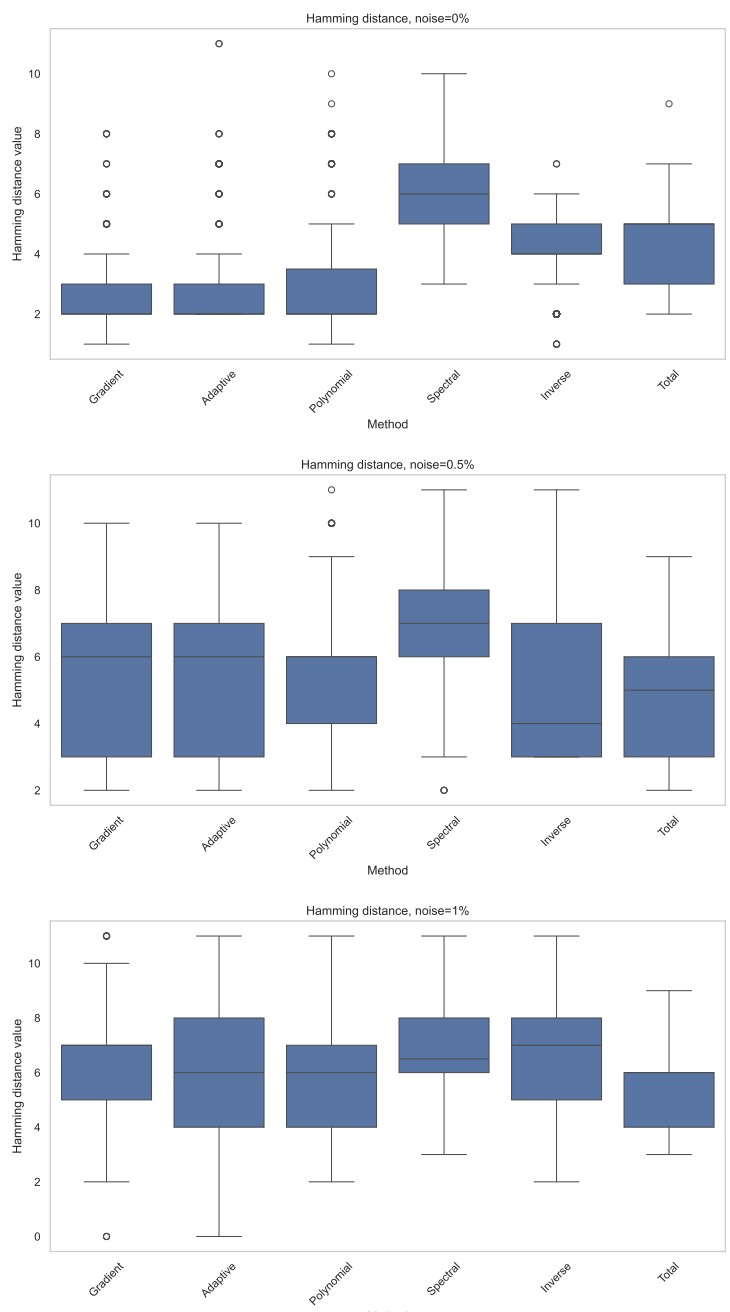

Figure 4: Distribution of coefficients values for different noise level

Table 10: Coefficients values calculated with EPDE, noise =0

| Methods/Terms | du/dt | d^3u/dx^3 | u*du/dx |
|---|---|---|---|
| Gradient | -0.4565 ± 0.2045 | 0.0008 ± 0.0038 | -1.3444 ± 0.1839 |
| Adaptive | -0.5045 ± 0.4228 | - | -0.0303 ± 0.0011 |
| Polynomial | -0.5045 ± 0.4228 | - | -0.0303 ± 0.0011 |
| Spectral | 0.0202 ± 0.0401 | 0.0002 ± 0.0000 | -0.2297 ± 0.1165 |
| Inverse | 0.0142 ± 0.0007 | - | -0.2412 ± 0.0004 |
| Total | -0.8334 ± 0.4282 | 1.1503 | -0.9770 ± 0.0231 |
| Ground truth | 1 | 1 | 6 |

Table 11: Coefficients values calculated with EPDE, noise =0.5

| Methods/Terms | du/dt | d^3u/dx^3 | u*du/dx |
|---|---|---|---|
| Gradient | -0.1973 ± 0.2523 | -0.0001 | -1.5692 ± 0.1447 |
| Adaptive | -0.6169 ± 0.2999 | - | -0.0292 ± 0.0016 |
| Polynomial | -0.9822 ± 0.0248 | -0.0022 | -0.8874 ± 0.2106 |
| Spectral | 0.0191 ± 0.0614 | -0.0000 ± 0.0003 | -0.1706 ± 0.1092 |
| Inverse | -0.4698 ± 0.1770 | 0.0001 ± 0.0003 | 0.0419 ± 0.0954 |
| Total | -0.8164 ± 0.1561 | 1.1569 ± 0.1997 | -0.9282 ± 0.0466 |
| Ground truth | 1 | 1 | 6 |

Table 12: Coefficients values calculated with EPDE, noise =1

| Methods/Terms | du/dt | d^3u/dx^3 | u*du/dx |
|---|---|---|---|
| Gradient | -0.3711 ± 0.1267 | -0.0819 ± 0.0647 | -0.7199 ± 0.3075 |
| Adaptive | -0.2784 ± 0.1686 | -0.0028 | -0.0953 ± 0.0751 |
| Polynomial | -0.6898 ± 0.1512 | -0.1111 ± 1.4098 | -0.9766 ± 0.1917 |
| Spectral | 0.0320 ± 0.0667 | -0.0001 ± 0.0006 | -0.1036 ± 0.0779 |
| Inverse | -0.1240 ± 0.0939 | 0.1313 ± 0.1499 | 0.0565 ± 0.0776 |
| Total | -0.8394 ± 0.1006 | - | -0.6780 ± 0.1103 |
| Ground truth | 1 | 1 | 6 |

Table 13: Coefficients values calculated with SINDy, noise =0

| Methods/Terms | du/dt | d^3u/dx^3 | u*du/dx |
|---|---|---|---|
| Gradient | 1 | -0.009 | 0.077 |
| Adaptive | 1 | - | - |
| Polynomial | 1 | - | 0.595 |
| Spectral | 1 | -0.067 | 2.530 |
| Inverse | 1 | 0.072 | 0.025 |
| Total | 1 | -4.011 | -0.599 |
| Ground truth | 1 | 1 | 6 |

Table 14: Coefficients values calculated with SINDy, noise =0.5

| Methods/Terms | du/dt | d^3u/dx^3 | u*du/dx |
|---|---|---|---|
| Gradient | 1 | 0.158 | 1.295 |
| Adaptive | 1 | - | - |
| Polynomial | 1 | - | 0.472 |
| Spectral | 1 | -0.066 | 2.530 |
| Inverse | 1 | - | - |
| Total | 1 | -4.010 | -0.639 |
| Ground truth | 1 | 1 | 6 |

Table 15: Coefficients values calculated with SINDy, noise =1

| Methods/Terms | du/dt | d^3u/dx^3 | u*du/dx |
|---|---|---|---|
| Gradient | 1 | 0.052 | 0.443 |
| Adaptive | 1 | - | - |
| Polynomial | 1 | - | 0.841 |
| Spectral | 1 | -0.067 | 2.523 |
| Inverse | 1 | - | - |
| Total | 1 | -4.015 | -0.731 |
| Ground truth | 1 | 1 | 6 |

# D  BURGERS EQUATION COEFFICIENTS AND DIFFERENTIATION ERRORS

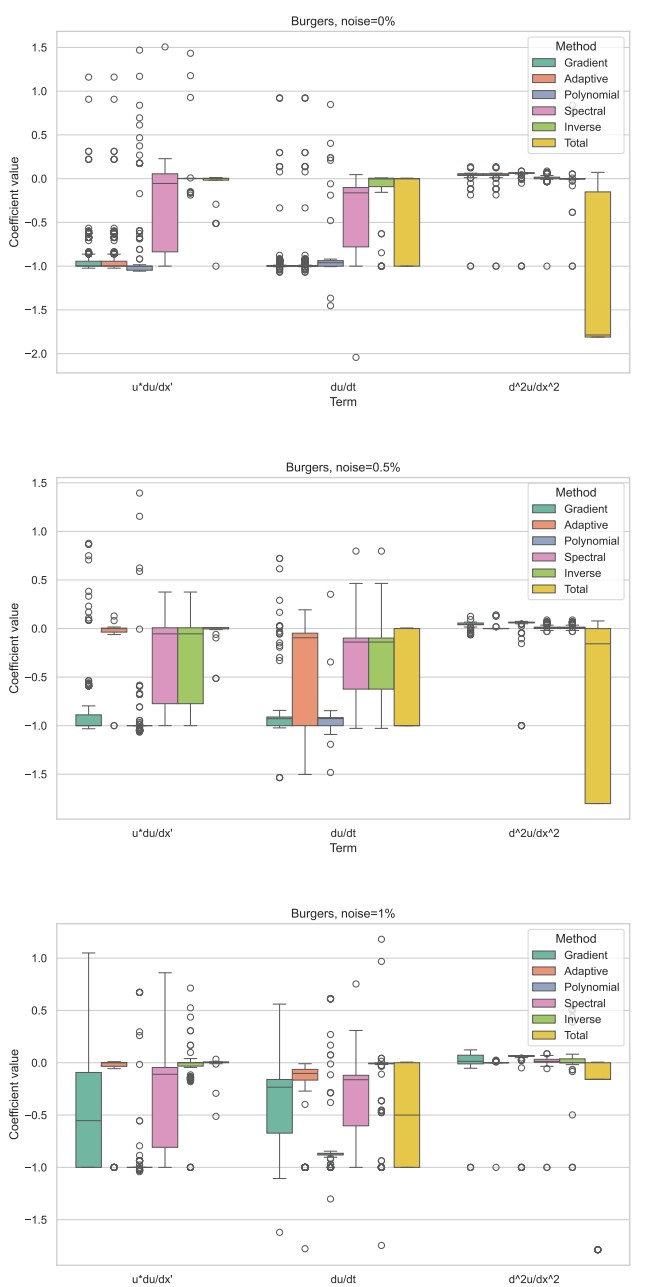

Figure 5: Distribution of coefficients values for different noise level

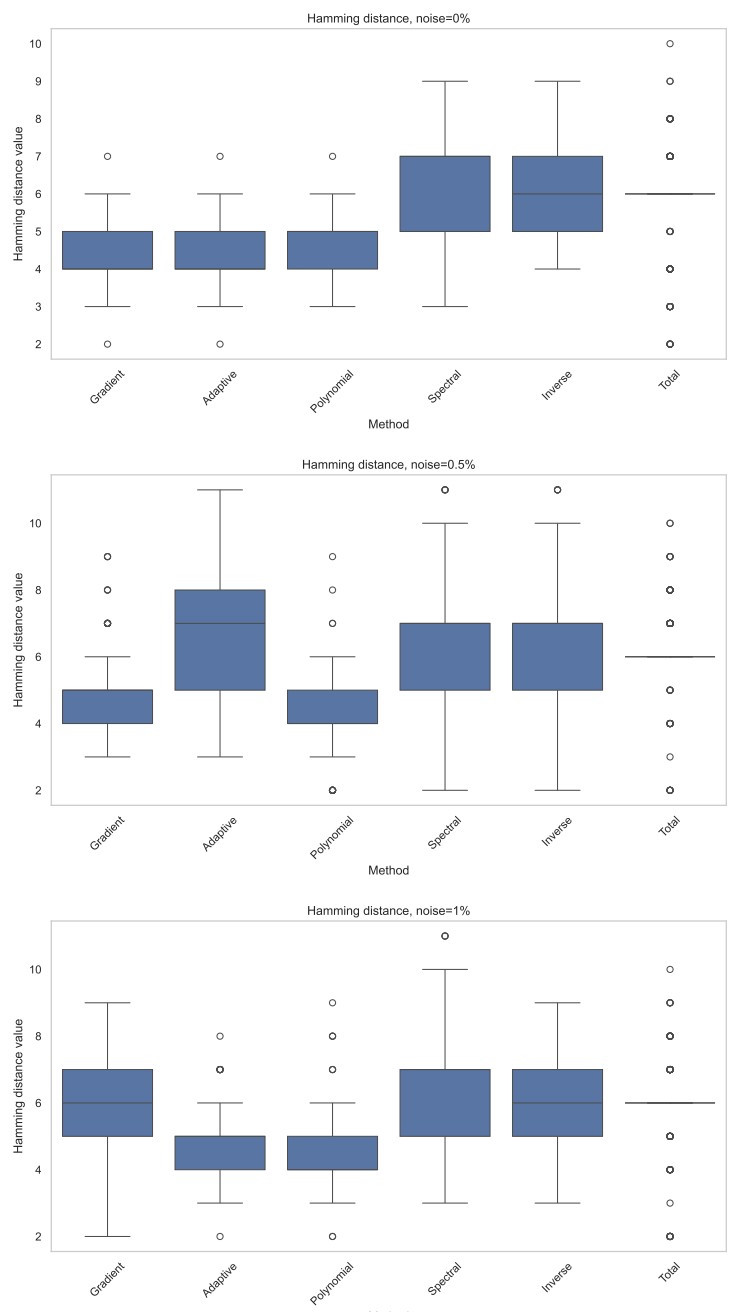

Figure 6: Distribution of coefficients values for different noise level

Table 16: Coefficients values calculated with EPDE, noise =0

| Methods/Terms | du/dt | d^2u/dx^2 | u*du/dx |
|---|---|---|---|
| Gradient | -0.9454 ± 0.0301 | 0.0346 ± 0.0133 | -0.8945 ± 0.0327 |
| Adaptive | -0.9454 ± 0.0301 | 0.0346 ± 0.0133 | -0.8945 ± 0.0327 |
| Polynomial | -0.9283 ± 0.0332 | 0.0439 ± 0.0188 | -0.8931 ± 0.0556 |
| Spectral | -0.4025 ± 0.0549 | 0.0032 ± 0.0171 | -0.3732 ± 0.0849 |
| Inverse | -0.2118 ± 0.1229 | -0.0239 ± 0.1173 | 0.0773 ± 0.1174 |
| Total | -0.4373 ± 0.2731 | -1.2180 ± 0.2525 | -0.1324 ± 0.1249 |
| Ground truth | 1 | -0.05 | 1 |

Table 17: Coefficients values calculated with EPDE, noise =0.5

| Methods/Terms | du/dt | d^2u/dx^2 | u*du/dx |
|---|---|---|---|
| Gradient | -0.8727 ± 0.0398 | 0.0428 ± 0.0035 | -0.8520 ± 0.0479 |
| Adaptive | -0.3827 ± 0.0621 | 0.0081 ± 0.0085 | -0.0278 ± 0.0198 |
| Polynomial | -0.9428 ± 0.0150 | 0.0378 ± 0.0202 | -0.9510 ± 0.0382 |
| Spectral | -0.3064 ± 0.0521 | 0.0138 ± 0.0033 | -0.3226 ± 0.0810 |
| Inverse | -0.3064 ± 0.0521 | 0.0138 ± 0.0033 | -0.3226 ± 0.0810 |
| Total | -0.5372 ± 0.3144 | -0.6960 ± 0.3206 | -0.0470 ± 0.0646 |
| Ground truth | 1 | -0.05 | 1 |

Table 18: Coefficients values calculated with EPDE, noise =1

| Methods/Terms | du/dt | d^2u/dx^2 | u*du/dx |
|---|---|---|---|
| Gradient | -0.4088 ± 0.0448 | 0.0051 ± 0.0269 | -0.5262 ± 0.0840 |
| Adaptive | -0.2707 ± 0.0511 | -0.0140 ± 0.0313 | -0.0913 ± 0.0478 |
| Polynomial | -0.8245 ± 0.0360 | 0.0384 ± 0.0208 | -0.9395 ± 0.0375 |
| Spectral | -0.3414 ± 0.0472 | 0.0049 ± 0.0207 | -0.3910 ± 0.0798 |
| Inverse | -0.1569 ± 0.0575 | 0.0238 ± 0.0420 | -0.0533 ± 0.0453 |
| Total | -0.4989 ± 0.1903 | -0.3595 ± 0.2082 | -0.0269 ± 0.0465 |
| Ground truth | 1 | -0.05 | 1 |

Table 19: Coefficients values calculated with SINDy, noise =0

| Methods/Terms | du/dt | d^2u/dx^2 | u*du/dx |
|---|---|---|---|
| Gradient | 1 | -0.044 | 0.952 |
| Adaptive | 1 | - | 0.041 |
| Polynomial | 1 | -0.058 | 1.057 |
| Spectral | 1 | - | 0.273 |
| Inverse | 1 | -0.134 | 0.205 |
| Total | 1 | 1.765 | - |
| Ground truth | 1 | -0.05 | 1 |

Table 20: Coefficients values calculated with SINDy, noise =0.5

| Methods/Terms | du/dt | d^2u/dx^2 | u*du/dx |
|---|---|---|---|
| Gradient | 1 | -0.044 | 0.955 |
| Adaptive | 1 | - | 0.041 |
| Polynomial | 1 | -0.055 | 1.039 |
| Spectral | 1 | - | 0.277 |
| Inverse | 1 | - | 0.188 |
| Total | 1 | 1.763 | - |
| Ground truth | 1 | -0.05 | 1 |

Table 21: Coefficients values calculated with SINDy, noise =1

| Methods/Terms | du/dt | d^2u/dx^2 | u*du/dx |
|---|---|---|---|
| Gradient | 1 | - | 0.661 |
| Adaptive | 1 | - | 0.041 |
| Polynomial | 1 | -0.050 | 1.004 |
| Spectral | 1 | - | 0.271 |
| Inverse | 1 | -0.129 | 0.202 |
| Total | 1 | 1.736 | -0.014 |
| Ground truth | 1 | -0.05 | 1 |

# E    WAVE EQUATION COEFFICIENTS AND DIFFERENTIATION ERRORS

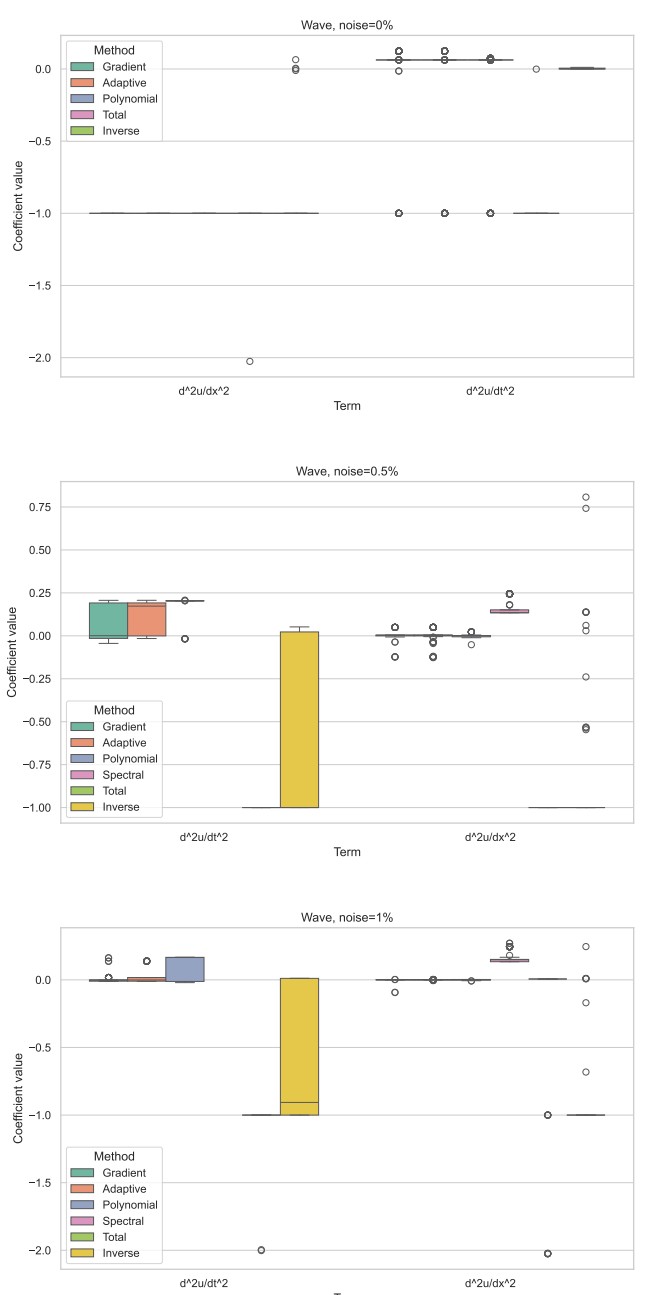

Figure 7: Distribution of coefficients values for different noise level

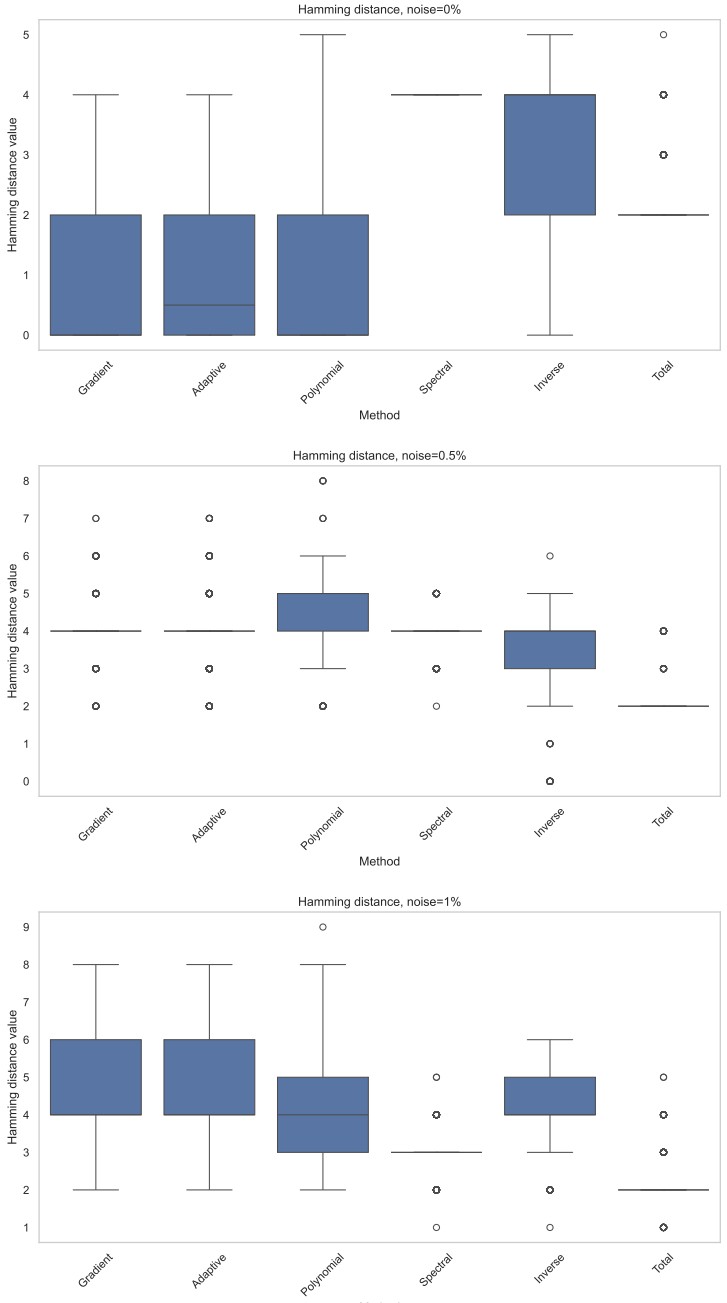

Figure 8: Distribution of coefficients values for different noise level

Table 22: Coefficients values calculated with EPDE, noise =0

| Methods/Terms | d^2u/dx^2 | d^2u/dt^2 |
|---|---|---|
| Gradient | -1 | -0.0005 ± 0.0338 |
| Adaptive | -1 | -0.0579 ± 0.0408 |
| Polynomial | -1 | -0.0827 ± 0.0430 |
| Spectral | - | - |
| Inverse | -0.9486 ± 0.0502 | 0.0038 ± 0.0162 |
| Total | -1.0049 ± 0.0097 | -0.9904 ± 0.0191 |
| Ground truth | 1 | -0.0625 |

Table 23: Coefficients values calculated with EPDE, noise =0.5

| Methods/Terms | d^2u/dx^2 | d^2u/dt^2 |
|---|---|---|
| Gradient | 0.0037 ± 0.0075 | 0.0688 ± 0.0199 |
| Adaptive | 0.0008 ± 0.0061 | 0.0972 ± 0.0131 |
| Polynomial | 0.0018 ± 0.0030 | 0.1539 ± 0.0345 |
| Spectral | 0.1549 ± 0.0041 | - |
| Inverse | -0.8171 ± 0.1025 | -0.5975 ± 0.1638 |
| Total | -1 | -1 |
| Ground truth | 1 | -0.0625 |

Table 24: Coefficients values calculated with EPDE, noise =1

| Methods/Terms | d^2u/dx^2 | d^2u/dt^2 |
|---|---|---|
| Gradient | -0.0041 ± 0.0068 | 0.0064 ± 0.0070 |
| Adaptive | 0.0004 ± 0.0003 | 0.0108 ± 0.0056 |
| Polynomial | -0.0001 ± 0.0005 | 0.1041 ± 0.0191 |
| Spectral | 0.1533 ± 0.0039 | - |
| Inverse | -0.8946 ± 0.0792 | -0.6434 ± 0.1901 |
| Total | -0.3441 ± 0.1927 | -1.0066 ± 0.0092 |
| Ground truth | 1 | -0.0625 |

Table 25: Coefficients values calculated with SINDy, noise =0

| Methods/Terms | d^2u/dx^2 | d^2u/dt^2 |
|---|---|---|
| Adaptive | 1 | -0.055 |
| Polynomial | 1 | -0.063 |
| Spectral | 1 | - |
| Inverse | 1 | -0.008 |
| Total | 1 | -0.007 |
| Ground truth | 1 | -0.0625 |

Table 26: Coefficients values calculated with SINDy, noise =0.5

| Methods/Terms | d^2u/dx^2 | d^2u/dt^2 |
|---|---|---|
| Gradient | 1 | -0.193 |
| Adaptive | 1 | -0.163 |
| Polynomial | 1 | -0.049 |
| Spectral | 1 | - |
| Inverse | 1 | -0.221 |
| Total | 1 | -0.027 |
| Ground truth | 1 | -0.0625 |

Table 27: Coefficients values calculated with SINDy, noise =1

| Methods/Terms | d^2u/dx^2 | d^2u/dt^2 |
|---|---|---|
| Gradient | 1 | -0.395 |
| Adaptive | 1 | -0.332 |
| Polynomial | 1 | -0.1 |
| Spectral | 1 | - |
| Inverse | 1 | -4.586 |
| Total | 1 | -0.079 |
| Ground truth | 1 | -0.0625 |

# F LAPLACE EQUATION COEFFICIENTS AND DIFFERENTIATION ERRORS

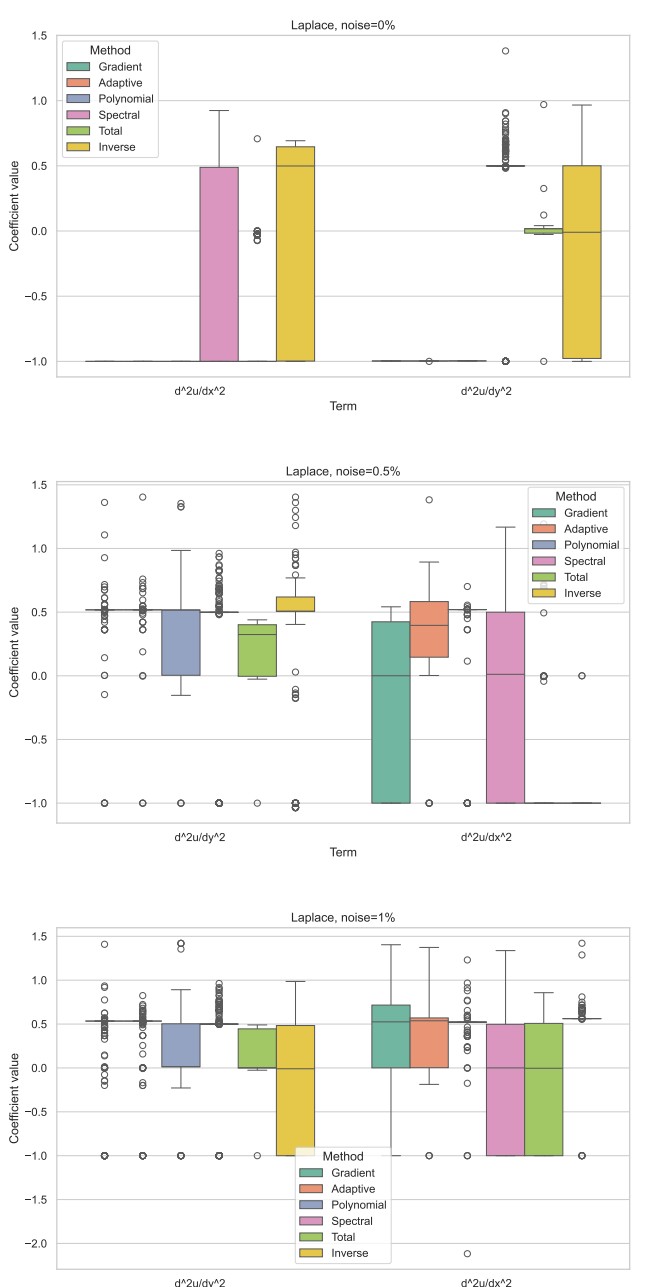

Figure 9: Distribution of coefficients values for different noise level

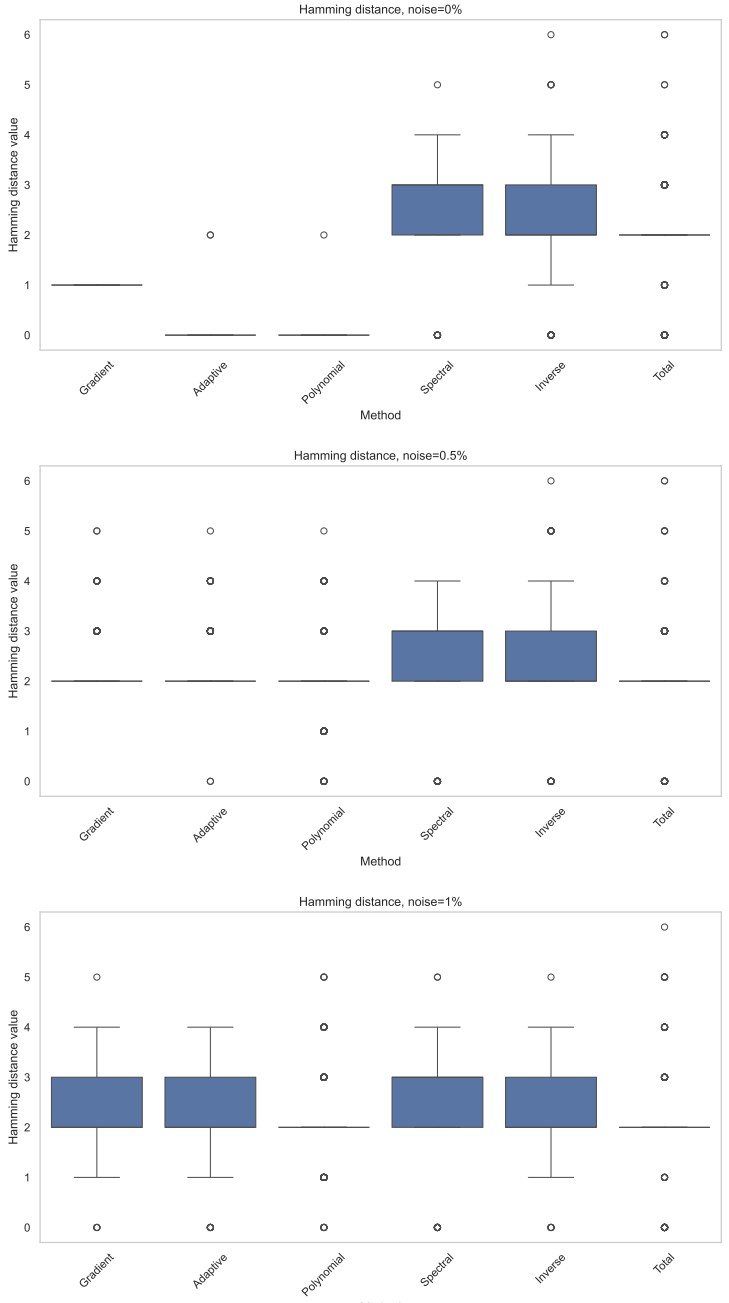

Figure 10: Distribution of coefficients values for different noise level

Table 28: Coefficients values calculated with EPDE, noise =0

| Methods/Terms | d^2u/dx^2 | d^2u/dy^2 |
|---|---|---|
| Gradient | -1 | -0.997 |
| Adaptive | -1 | -0.997 |
| Polynomial | -1 | -0.9964 |
| Spectral | -1 | -1 |
| Inverse | -0.9985 ± 0.0007 | -0.9955 ± 0.0013 |
| Total | -1 | -1 |
| Ground truth | 1 | 1 |

Table 29: Coefficients values calculated with EPDE, noise =0.5

| Methods/Terms | d^2u/dx^2 | d^2u/dy^2 |
|---|---|---|
| Gradient | 0.2072 ± 0.1540 | 0.5129 ± 0.0095 |
| Adaptive | 0.4755 ± 0.1158 | 0.5150 ± 0.0065 |
| Polynomial | 0.5139 ± 0.0077 | 0.3348 ± 0.0365 |
| Spectral | 0.4728 ± 0.0783 | 0.5393 ± 0.0137 |
| Inverse | -0.0000 ± 0.0025 | 0.5579 ± 0.0239 |
| Total | 0.3411 ± 0.1787 | 0.1874 ± 0.0382 |
| Ground truth | 1 | 1 |

Table 30: Coefficients values calculated with EPDE, noise =1

| Methods/Terms | d^2u/dx^2 | d^2u/dy^2 |
|---|---|---|
| Gradient | 0.4476 ± 0.1022 | 0.4962 ± 0.0192 |
| Adaptive | 0.4047 ± 0.0796 | 0.5056 ± 0.0164 |
| Polynomial | 0.5100 ± 0.0140 | 0.2119 ± 0.0521 |
| Spectral | 0.3388 ± 0.1160 | 0.5333 ± 0.0117 |
| Inverse | 0.5868 ± 0.0061 | 0.3661 ± 0.0714 |
| Total | 0.3295 ± 0.0528 | 0.2150 ± 0.0566 |
| Ground truth | 1 | 1 |

Table 31: Coefficients values calculated with SINDy, noise =0

| Methods/Terms | d^2u/dx^2 | d^2u/dy^2 |
|---|---|---|
| Gradient | 1 | 1.028 |
| Adaptive | 1 | 1.129 |
| Polynomial | 1 | 1.009 |
| Spectral | 1 | - |
| Inverse | 1 | 0.62 |
| Total | 1 | - |
| Ground truth | 1 | 1 |

Table 32: Coefficients values calculated with SINDy, noise =0.5

| Methods/Terms | d^2u/dx^2 | d^2u/dy^2 |
|---|---|---|
| Gradient | 1 | - |
| Adaptive | 1 | - |
| Polynomial | 1 | - |
| Spectral | 1 | - |
| Inverse | 1 | - |
| Total | 1 | - |
| Ground truth | 1 | 1 |

Table 33: Coefficients values calculated with SINDy, noise =1

| Methods/Terms | d^2u/dx^2 | d^2u/dy^2 |
|---|---|---|
| Gradient | 1 | - |
| Adaptive | 1 | -0.457 |
| Polynomial | 1 | - |
| Spectral | 1 | - |
| Inverse | 1 | - |
| Total | 1 | - |
| Ground truth | 1 | 1 |

# G    QUASIGEOSTROPHIC POTENTIAL VORTICITY EQUATION

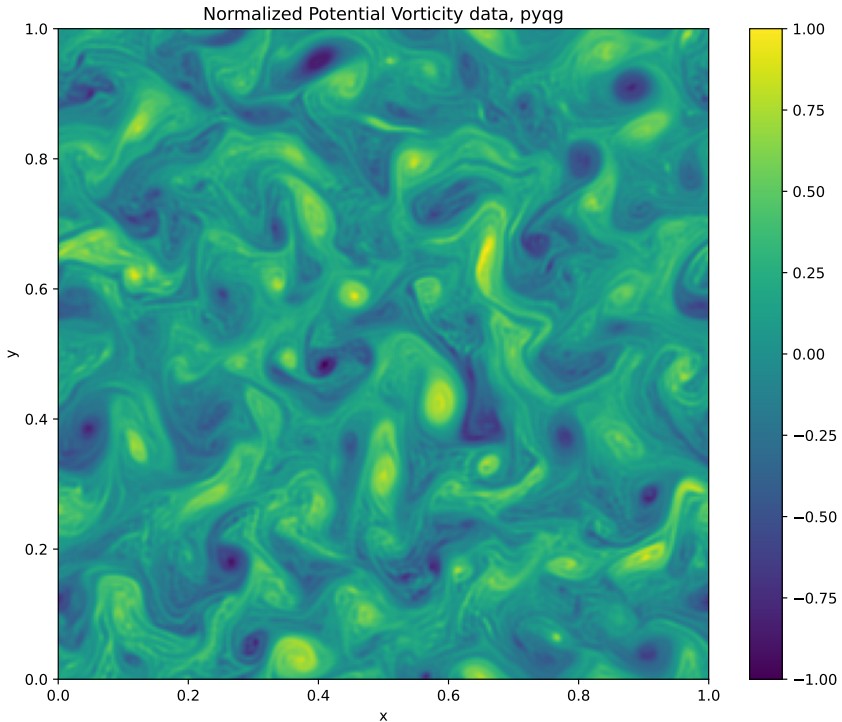

Figure 11: Normalized potential vorticity data, pyqg

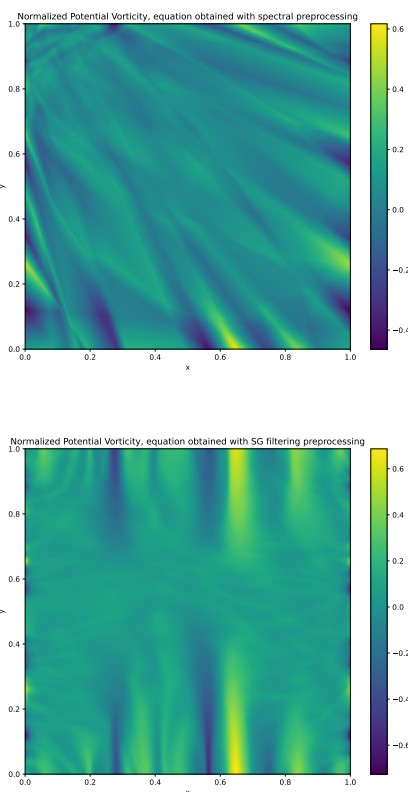

Figure 12: Normalized Potential Vorticity, equation obtained with spectral preprocessing(left) and SG filtering preprocessing(right)

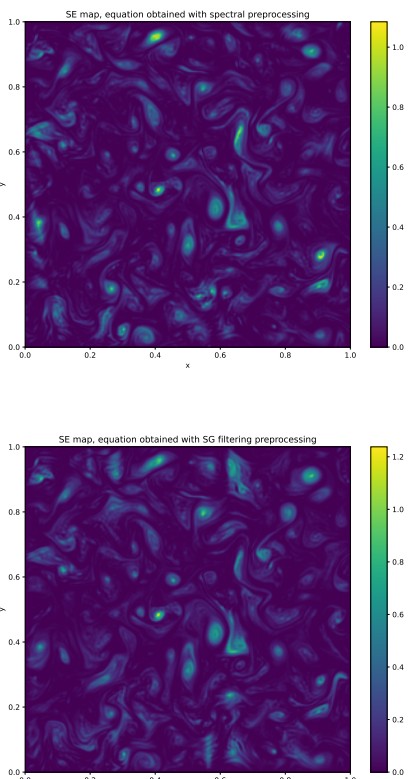

Figure 13: SE map, equation obtained with spectral preprocessing (left, MSE = 0.057) and SG filtering preprocessing(right, MSE = 0.065)

# H HAMMING STRUCTURAL DISTANCES

Table 34: SHD for equations calculated with EPDE, noise = 0

| Methods/Equations | Burgers | KdV | Laplace | ode | Wave |
|---|---|---|---|---|---|
| Gradient | 4 ± 0.091 | 3 ± 0.1302 | 1 | 0 + 0.0491 | 1 ± 0.1206 |
| Adaptive | 4 ± 0.091 | 3 ± 0.1655 | 0 + 0.0159 | 0 + 0.0636 | 1 ± 0.1189 |
| Polynomial | 4 ± 0.272 | 3 ± 0.1824 | 0 + 0.0112 | 0 + 0.0654 | 1 ± 0.119 |
| Spectral | 6 ± 0.1993 | 6 ± 0.1698 | 3 ± 0.0999 | 3 ± 0.1855 | 4 |
| Inverse | 6 ± 0.3554 | 4 ± 0.1267 | 2 ± 0.1432 | 4 ± 0.2077 | 3 ± 0.1314 |
| Total | 6 ± 0.1299 | 4 ± 0.1332 | 2 ± 0.1169 | 0 ± 0.0749 | 2 ± 0.0726 |

Table 35: SHD for equations calculated with EPDE, noise = 0.5

| Methods/Equations | Burgers | KdV | Laplace | ode | Wave |
|---|---|---|---|---|---|
| Gradient | 5 ± 0.1412 | 5 ± 0.2209 | 2 ± 0.0629 | 0 + 0.0535 | 4 ± 0.0979 |
| Adaptive | 7 ± 0.2144 | 5 ± 0.2836 | 2 ± 0.0637 | 0 + 0.0458 | 4 ± 0.0671 |
| Polynomial | 4 ± 0.1334 | 5 ± 0.1935 | 2 ± 0.1062 | 0 + 0.0734 | 4 ± 0.1237 |
| Spectral | 6 ± 0.2107 | 7 ± 0.2087 | 3 ± 0.0992 | 3 ± 0.1971 | 4 ± 0.0631 |
| Inverse | 6 ± 0.2107 | 5 ± 0.2632 | 3 ± 0.124 | 4 ± 0.1531 | 3 ± 0.1305 |
| Total | 6 ± 0.1152 | 5 ± 0.1394 | 2 ± 0.1084 | 0 + 0.0783 | 2 ± 0.0571 |

Table 36: SHD for equations calculated with EPDE, noise = 1

| Methods/Equations | Burgers | KdV | Laplace | ode | Wave |
|---|---|---|---|---|---|
| Gradient | 6 ± 0.2134 | 6 ± 0.2824 | 2 ± 0.0843 | 0 + 0.0424 | 5 ± 0.1186 |
| Adaptive | 4 ± 0.127 | 6 ± 0.3199 | 2 ± 0.0815 | 0 + 0.0433 | 5 ± 0.0833 |
| Polynomial | 4 ± 0.1248 | 6 ± 0.247 | 2 ± 0.102 | 0 + 0.0627 | 4 ± 0.1252 |
| Spectral | 6 ± 0.2096 | 7 ± 0.1827 | 3 ± 0.0974 | 3 ± 0.1904 | 3 ± 0.0687 |
| Inverse | 6 ± 0.2057 | 7 ± 0.2509 | 2 ± 0.0883 | 4 ± 0.1231 | 4 ± 0.1004 |
| Total | 6 ± 0.1283 | 5 ± 0.133 | 2 ± 0.1064 | 0 ± 0.0792 | 2 ± 0.0847 |

Table 37: SHD for equations calculated with SINDy, noise = 0

| Methods/Equations | Burgers | KdV | Laplace | ode | Wave |
|---|---|---|---|---|---|
| Gradient | 0 | 1 | 0 | 0 | 0 |
| Adaptive | 1 | 3 | 0 | 0 | 1 |
| Polynomial | 1 | 3 | 0 | 0 | 1 |
| Spectral | 1 | 2 | 1 | 1 | 1 |
| Inverse | 1 | 1 | 1 | 0 | 1 |
| Total | 5 | 4 | 2 | 0 | 2 |

Table 38: SHD for equations calculated with SINDy, noise = 0.5

| Methods/Equations | Burgers | KdV | Laplace | ode | Wave |
|---|---|---|---|---|---|
| Gradient | 0 | 2 | 2 | 0 | 2 |
| Adaptive | 1 | 3 | 2 | 0 | 2 |
| Polynomial | 1 | 2 | 2 | 0 | 2 |
| Spectral | 1 | 2 | 2 | 1 | 1 |
| Inverse | 3 | 3 | 4 | 0 | 1 |
| Total | 5 | 4 | 2 | 0 | 2 |

Table 39: SHD for equations calculated with SINDy, noise = 1

| Methods/Equations | Burgers | KdV | Laplace | ode | Wave |
|---|---|---|---|---|---|
| Gradient | 2 | 2 | 3 | 0 | 2 |
| Adaptive | 1 | 3 | 1 | 0 | 2 |
| Polynomial | 0 | 5 | 4 | 0 | 2 |
| Spectral | 1 | 2 | 2 | 1 | 1 |
| Inverse | 1 | 4 | 4 | 0 | 2 |
| Total | 3 | 4 | 2 | 0 | 2 |

# I DIFFERENTIATION ERRORS

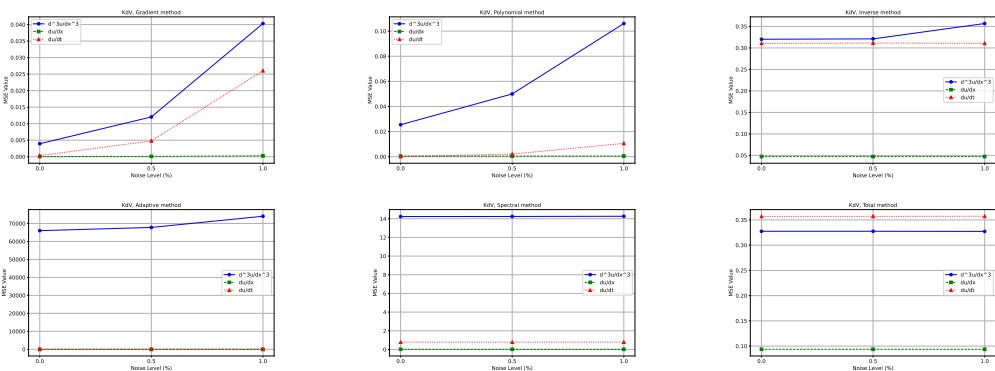

Figure 14: Differentiation errors (MSE) for KdV equation with different noise level

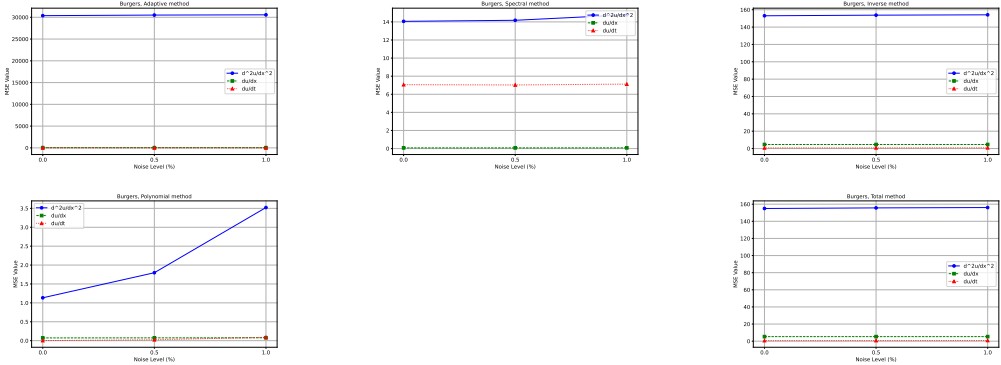

Figure 15: Differentiation errors (MSE) for Burgers equation with different noise level

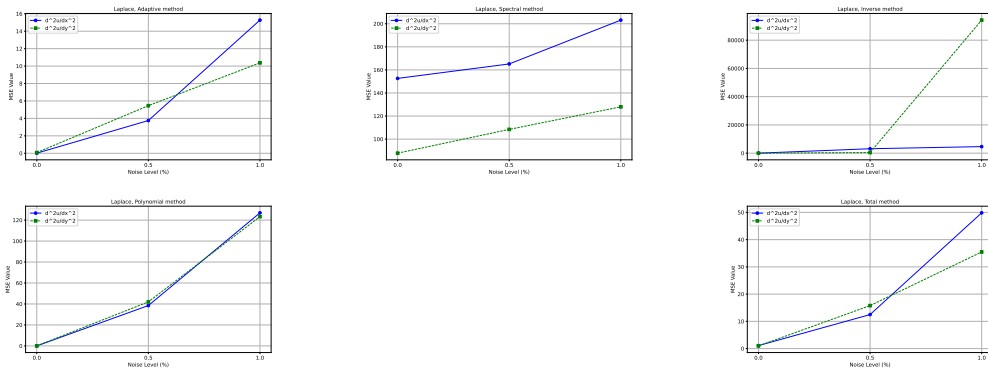

Figure 16: Differentiation errors (MSE) for Laplace equation with different noise level

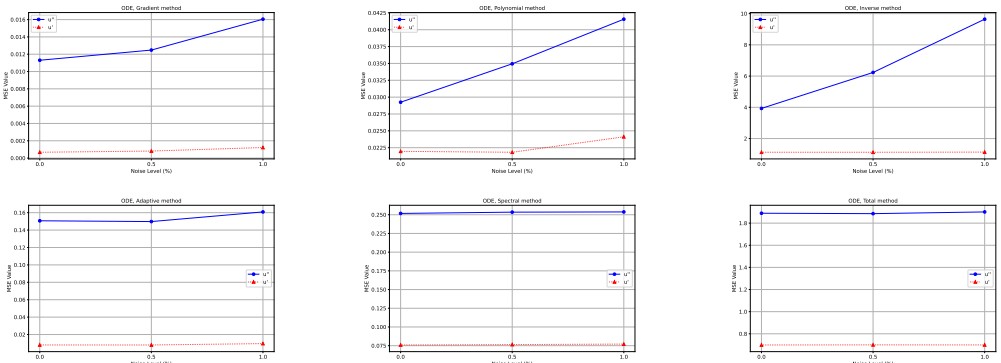

Figure 17: Differentiation errors (MSE) for ODE equation with different noise level

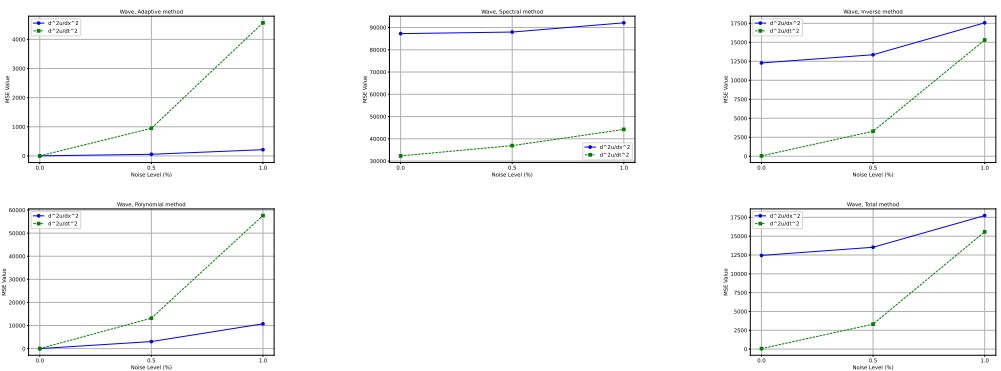

Figure 18: Differentiation errors (MSE) for Wave equation with different noise level

Table 40: Differentiation errors, noise = 0

| Methods/Terms | Burgers equation | | |
|---|---|---|---|
| | du/dt | d^2u/dx^2 | du/dx |
| Adaptive | 0.4530 | 30374.7349 | 43.9103 |
| Polynomial | 0.0012 | 1.1339 | 0.0724 |
| Spectral | 7.0597 | 14.0607 | 0.0757 |
| Inverse | 0.7638 | 153.0405 | 4.6508 |
| Total | 0.6760 | 154.9538 | 5.3229 |
| Methods/Terms | KdV equation | | |
| | du/dt | d^3u/dx^3 | du/dx |
| Gradient | 0.000329 | 0.0039 | 0.00004184 |
| Adaptive | 0.2540 | 65999.0757 | 4.0918 |
| Polynomial | 0.00007761 | 0.0254 | 0.0004532 |
| Spectral | 0.7798 | 14.2334 | 0.0129 |
| Inverse | 0.3105 | 0.3202 | 0.0473 |
| Total | 0.357 | 0.3275 | 0.0937 |
| Methods/Terms | Laplace equation | | |
| | d^2u/dx^2 | d^2u/dy^2 | |
| Adaptive | 0.0124 | 0.0814 | |
| Polynomial | 0.0058 | 0.0036 | |
| Spectral | 152.6378 | 87.813 | |
| Inverse | 0.0098 | 0.178 | |
| Total | 1.0941 | 0.9853 | |
| Methods/Terms | ODE equation | | |
| | u' | u'' | |
| Gradient | 0.00067 | 0.0113 | |
| Adaptive | 0.00801 | 0.1507 | |
| Polynomial | 0.022 | 0.0292 | |
| Spectral | 0.0759 | 0.2517 | |
| Inverse | 1.1231 | 3.9290 | |
| Total | 0.6988 | 1.8895 | |
| Methods/Terms | Wave equation | | |
| | d^2u/dx^2 | d^2u/dt^2 | |
| Adaptive | 2.0602 | 1.2876 | |
| Polynomial | 9.8417 | 0.0057 | |
| Spectral | 87251.4516 | 32328.7991 | |
| Inverse | 12282.1612 | 47.9087 | |
| Total | 12444.1383 | 65.6059 | |

Table 41: Differentiation errors, noise = 0.5

| | Burgers equation | | |
|---|---|---|---|
| Methods/Terms | du/dt | d^2u/dx^2 | du/dx |
| Adaptive | 0.478 | 30505.6170 | 43.9511 |
| Polynomial | 0.0234 | 1.7973 | 0.0737 |
| Spectral | 7.0293 | 14.1771 | 0.0761 |
| Inverse | 0.8013 | 153.7448 | 4.6542 |
| Total | 0.7139 | 155.6498 | 5.3271 |
| | KdV equation | | |
| Methods/Terms | du/dt | d^3u/dx^3 | du/dx |
| Gradient | 0.00478 | 0.0120 | 0.0000947 |
| Adaptive | 0.2544 | 67786.0505 | 4.0969 |
| Polynomial | 0.0021 | 0.05 | 0.0004592 |
| Spectral | 0.7804 | 14.2418 | 0.0128 |
| Inverse | 0.3114 | 0.3211 | 0.0473 |
| Total | 0.3571 | 0.3275 | 0.0937 |
| | Laplace equation | | |
| Methods/Terms | d^2u/dx^2 | d^2u/dy^2 | |
| Adaptive | 3.7555 | 5.4563 | |
| Polynomial | 38.4966 | 41.9891 | |
| Spectral | 165.1737 | 108.4259 | |
| Inverse | 3138.8938 | 426.1978 | |
| Total | 12.4789 | 15.7744 | |
| | ODE equation | | |
| Methods/Terms | u' | u'' | |
| Gradient | 0.00081 | 0.0124 | |
| Adaptive | 0.008 | 0.1498 | |
| Polynomial | 0.0218 | 0.0349 | |
| Spectral | 0.0764 | 0.2534 | |
| Inverse | 1.1220 | 6.2318 | |
| Total | 0.6998 | 1.8857 | |
| | Wave equation | | |
| Methods/Terms | d^2u/dx^2 | d^2u/dt^2 | |
| Adaptive | 56.2421 | 947.3597 | |
| Polynomial | 3047.6637 | 13167.1760 | |
| Spectral | 87963.7107 | 36937.9071 | |
| Inverse | 13359.6413 | 3287.3561 | |
| Total | 13524.8049 | 3315.1318 | |

Table 42: Differentiation errors, noise = 1

| | Burgers equation | | |
|---|---|---|---|
| Methods/Terms | du/dt | d^2u/dx^2 | du/dx |
| Adaptive | 0.5383 | 30565.1709 | 43.9511 |
| Polynomial | 0.088 | 3.5219 | 0.0761 |
| Spectral | 7.1244 | 14.7814 | 0.0783 |
| Inverse | 0.8933 | 154.1399 | 4.6562 |
| Total | 0.8085 | 156.0558 | 5.3286 |
| | KdV equation | | |
| Methods/Terms | du/dt | d^3u/dx^3 | du/dx |
| Gradient | 0.0260 | 0.0403 | 0.0002695 |
| Adaptive | 0.2543 | 74009.2666 | 4.1105 |
| Polynomial | 0.0106 | 0.1061 | 0.000499 |
| Spectral | 0.7811 | 14.2637 | 0.0129 |
| Inverse | 0.3108 | 0.3568 | 0.0473 |
| Total | 0.3573 | 0.3273 | 0.0936 |
| | Laplace equation | | |
| Methods/Terms | d^2u/dx^2 | d^2u/dy^2 | |
| Adaptive | 15.2715 | 10.3725 | |
| Polynomial | 126.9283 | 123.2955 | |
| Spectral | 203.1654 | 128.0034 | |
| Inverse | 4647.4401 | 94205.9511 | |
| Total | 49.8341 | 35.4761 | |
| | ODE equation | | |
| Methods/Terms | u' | u'' | |
| Gradient | 0.00122 | 0.0160 | |
| Adaptive | 0.0096 | 0.1609 | |
| Polynomial | 0.0241 | 0.0416 | |
| Spectral | 0.0771 | 0.2538 | |
| Inverse | 1.132 | 9.6419 | |
| Total | 0.6993 | 1.9013 | |
| | Wave equation | | |
| Methods/Terms | d^2u/dx^2 | d^2u/dt^2 | |
| Adaptive | 213.6322 | 4567.2138 | |
| Polynomial | 10740.0809 | 57619.2452 | |
| Spectral | 92107.8232 | 44231.8631 | |
| Inverse | 17561.9236 | 15300.6987 | |
| Total | 17738.1271 | 15580.1558 | |

