# OpenReview forum: "Numerical Differentiation and Its Impact on Uncertainty in Learned Dynamical Systems"
_mathai.club/MathAI/2025/Conference — MathAI 2025 Oral_

### Official Review · Reviewer_nWRf · 2025-02-26
**"Numerical Differentiation and Its Impact on Uncertainty in Learned Dynamical Systems"**

**Rating:** 7
**Confidence:** 4

**Review:**

The article is pesented the important problem of the influence of numerical differentiation in the context of discovering differential equations from a set of experimental data.
The authors consistently explain why the quality of derivative calculations significantly affects the accuracy and reliability of the discovered equations, and consider several methods of numerical differentiation.

**Strengths**
- A comprehensive review of the issue of the influence of differentiation methods on the equation discovery procedure.
- A rich experimental section: the authors study several different differential equations, including both analytical examples and numerical solutions.
- Comparison of two different equation discovery frameworks (SINDy and EPDE) for different noise levels.
- Many different aspects related to the quality of the found equations are considered: differentiation errors, errors in coefficients, structural errors.

**Weaknesses**
- Closed formulation of experiments in terms of details: how hyperparameters were chosen is not always described in detail.
- Insufficiently developed practical guide to choosing a differentiation method: the results, although collected in overview tables, are not summarized in terms of specific conclusions.

**Problems with the presentation of the material.**
The article contains several inaccuracies and typos:
1. lines 109 and 177 - the index "j" is indicated, but the listing is by index "i".
2. line 130 - it is indicated "maybe link he", but the link is not provided.
3. Different notations are used to designate derivatives in the text of the article, for example, in line 315 the combination of symbols " ′” " is used for the third derivative.
4. Section "5 DISCUSSION" and Appendices B, C, D, E, F, H, I list six methods (Gradient, Adaptive, Polynomial, Spectral, Inverse, Total), but the paper itself discusses only three (Savitzky-Golay filtering, Spectral domain differentiation, Total variation regularization) or four (Finite difference, Savitzky-Golay filtering, Spectral domain differentiation, Total variation regularization).
5. Section "5 DISCUSSION" does not specify the error metric for differentiation error and coefficients error.
6. Lines 427 and 431 - the terms "lesser" and "second-best" are mentioned in relation to differentiation methods, but the order on the set of methods is not specified.
7. Line 631 - the value of cutoff frequency and the method for choosing this value are not specified.

**Recommendation.**
I recommend the article for publication provided that the authors address the above weaknesses and areas for improvement.

---

### Official Review · Reviewer_FZD6 · 2025-02-27
**Numerical Differentiation and Its Impact on Uncertainty in Learned Dynamical Systems**

**Rating:** 7
**Confidence:** 4

**Review:**

The article presents a comparative analysis of the impact of numerical differentiation on the quality of solutions to differential equations in the context of machine learning.

**Strengths:**

- A comprehensive list of various differentiation approaches is provided, with explanations of the workings and properties of the methods.
- The experimental section examines several differential equations with different levels of complexity. All experiments include both error tables, which allow for method comparison, and the distribution of coefficients with respect to noise.
- Essential derivations for understanding the topic are placed in the appendix, simplifying the overall comprehension of the article.

**Weaknesses:**

- The experiments are poorly described, with no information provided on hyperparameters or models. There is also a lack of information about the runtime and convergence of the methods. These aspects do not allow for the verification of the quality of the experiments conducted and make it impossible to replicate the results.
- Some abbreviations for method names in all presented tables do not clarify which specific method was applied. Abbreviations need to be replaced with full names.
- The article repeatedly mentions a variant of solving differential equations using the PINN architecture. Although this method is actively developing, there is either no comparison with it or it is described implicitly.

---

### Official Review · Reviewer_tmu1 · 2025-02-27
**Practical Aspects of Numerical Differentiation and Its Impact on Uncertainty in Learned Dynamical Systems**

**Rating:** 7
**Confidence:** 4

**Review:**

The paper investigates how numerical differentiation methods affect the discovery of differential equations from noisy data. It examines three main approaches: Savitzky–Golay (SG) filtering, spectral differentiation (including neural network–based approximation), and derivative regularization (via techniques such as total variation regularization). The experiments cover a range of test cases—from a simple second-order ODE to various PDEs (KdV, Burgers, wave equations) and even a real-data example using atmospheric modeling (pyqg). The performance is assessed using quantitative metrics like Structural Hamming Distance (SHD) and coefficient errors across different noise levels (0%, 50%, 100%).

The methodology is robust. SG filtering works by locally fitting polynomials to smooth the data and then differentiating analytically. Its performance depends critically on window size and polynomial order, which can either under- or over-smooth the data. Spectral methods, applied both directly (via FFT and Butterworth filtering) and through a neural network approximation that inherently prioritizes low-frequency (smooth) components, offer an alternative by effectively suppressing high-frequency noise. The paper also employs variational approaches that regularize the computed derivatives, which are particularly useful for reducing noise-induced errors.

A key strength of the paper is the experimental design. The authors test the differentiation methods on a diverse set of differential equations, ensuring that the conclusions are not limited to a single class of problems. Moreover, they utilize two independent algorithms for equation discovery—SINDy (a sparse regression approach) and EPDE (an evolutionary algorithm)—to verify that the influence of the differentiation method is consistent regardless of the subsequent identification technique. Multiple runs with statistical averaging (using boxplots and error metrics) further bolster the reliability of the results.

Despite these merits, some weaknesses are apparent. The selection of hyperparameters (e.g., the SG filter window, cutoff frequency in spectral filtering, and regularization parameters) is critical for performance, yet the paper does not offer a systematic strategy for their optimization. Additionally, while the empirical results are convincing, the paper lacks a deeper theoretical discussion that would explain, from first principles, why certain methods outperform others under specific noise conditions. Finally, although the chosen test cases are diverse, the study does not include highly chaotic systems or scenarios with irregular data sampling, which could limit the generalizability of the conclusions.

Strength:

Comprehensive Comparison: Evaluates multiple numerical differentiation techniques (SG, spectral, and regularization-based) under varied noise conditions.
Robust Experimental Design: Uses a wide range of test equations (from simple ODEs to complex PDEs) and two independent equation discovery methods (SINDy and EPDE).
Quantitative Evaluation: Employs clear performance metrics (SHD, coefficient error) and statistical averaging across multiple runs.
Practical Recommendations: Provides useful guidelines for selecting differentiation methods based on data quality and noise level.

Weaknesse:

Hyperparameter Sensitivity: The results depend on careful tuning of parameters, with limited guidance on optimal selection.
Limited Theoretical Analysis: Lacks an in-depth theoretical explanation for the observed performance differences among methods.
Scope of Test Cases: Does not consider chaotic systems or non-uniformly sampled data, which might affect the robustness of the findings.
Empirical Focus: While the empirical approach is strong, additional theoretical insights could further validate the conclusions.

---

### Official Review · Reviewer_FxCn · 2025-02-27
**A reasonable study, but the results are not clear to me**

**Rating:** 6
**Confidence:** 4

**Review:**

Contents of the paper
-----------------------------

The paper is an experimental study of how different numerical differentiation approaches affect the reconstruction of the differential equation from the data. The authors consider two frameworks for fitting the equations, SINDy and EPDE. They consider a range of methods to compute direct or filtered derivatives and test them on a range of examples (several classical PDEs and quasigeostrophic data). The performance results are presented in several tables.

Strengths
-------------

The paper systematically addresses the dependence of the ODE/PDE reconstruction on the differentiation method. The authors have performed a wide range of experiments across multiple methods and equations. Appendices contain detailed statistics for each equation. The authors have open-sourced their code. References include many recent publications, placing the study in a state-of-the-art context.

Weaknesses
-----------------

**Exposition.** While generally readable, the paper has issues with the exposition.

The paper includes lengthy discussions of topics that do not seem to be directly relevant to its main question (e.g. different kinds of uncertainty in the introduction), while being relatively vague or unclear in the central questions. In particular, I don't think that I have really understood the important frameworks SINDy and EPDE from their discussion in section 2. My understanding is that SINDy simply fits a sparse linear combination of predefined functions constructed from the solution and its derivatives, while EPDE additionally tries to construct new such functions, primarily by forming products. In the description of EPDE, the structure of the equation is confusingly intertwined with the optimization algorithm (evolutionary optimization) employed to find this structure.

Structural Hamming Distance (SHD) seems to play an important role in the assessment of results, but I don't see where in the paper this concept is explained. My guess is that this is the number of false plus missing terms found in the differential equation.

In the tables 1-3, the row names ("Gradient", etc.) are not explained. Does "Total" mean the sum of values in the column, or does it refer to the Total variation regularization?  What is "D. error", and why is there such a huge difference between different methods (Gradient: 0.003248, Adaptive: 8035.51)? This should be explained in the paper.

**Results.** Most importantly, I don't understand what are the results of this study. What specifically is the takeaway for the reader, apart from the obvious idea "more noise requires more filtering"?

I'm not even convinced that the methodology implemented in the paper produces useful results in the particular examined examples. Сonsider the Quasigeostrophic problem in section 4.7. The reconstructed equations are proposed in formulas (14), (15). How do we know that these equations are (more or less) correct? Suppose that I divide the domain into two parts and use the data from one part to reconstruct the equation. Ideally, I would expect that solving this equation in the other part of the domain would (approximately) match the other part of the data. Is that so in the case at hand? It seems not; the authors don't claim anything like that. Can we at least say that solving the equation in the training part matches the training data? It appears that even this is not true. The authors provide some pictures in appendix G, but do not explain what they mean. Can we then at least say that the equation reproduces typical patterns in the data, e.g., vortices, finite signal propagattion speed, etc.? I don't see any discussion of that. In summation, I don't see clear evidence that the proposed equations (14), (15) actually describe the data at hand. Similar criticism can be addressed at other examples, too.

One of the conclusion says: "Best differentiation methods for noise data and clean data are different."  This statement is very unspecific and, again, it is obviously anticipated that the utility of the methods with built-in noise filtering should grow with the amount of noise. Are there any more  specific insights resulting from this study? How are they implied by tables 1-3 and what is the rationale behind them?

---

### Decision · Program_Chairs · 2025-03-08

**Decision:**

Accept (Oral)

**Comment:**

Your article has been accepted and you can give a talk on the article. All articles will be sorted by rating and within the available conference places one author from each article will be invited. If there are not enough places, then you will either have the opportunity to speak remotely or come at your own expense!